# SPEX: Scaling Feature Interaction Explanations for LLMs

Justin Singh Kang [* 1]   Landon Butler [* 1]   Abhineet Agarwal [* 2]   Yigit Efe Erginbas [1]   Ramtin Pedarsani [3]
Bin Yu [1 2]   Kannan Ramchandran [1]

## Abstract

Large language models (LLMs) have revolutionized machine learning due to their ability to capture complex interactions between input features. Popular post-hoc explanation methods like SHAP provide *marginal* feature attributions, while their extensions to interaction importances only scale to small input lengths ($\approx 20$). We propose *Spectral Explainer* (SPEX), a model-agnostic interaction attribution algorithm that efficiently scales to large input lengths ($\approx 1000$). SPEX exploits underlying natural sparsity among interactions—common in real-world data—and applies a sparse Fourier transform using a channel decoding algorithm to efficiently identify important interactions. We perform experiments across three difficult long-context datasets that require LLMs to utilize interactions between inputs to complete the task. For large inputs, SPEX outperforms marginal attribution methods by up to 20% in terms of faithfully reconstructing LLM outputs. Further, SPEX successfully identifies key features and interactions that strongly influence model output. For one of our datasets, *HotpotQA*, SPEX provides interactions that align with human annotations. Finally, we use our model-agnostic approach to generate explanations to demonstrate abstract reasoning in closed-source LLMs (*GPT-4o mini*) and compositional reasoning in vision-language models.

## 1. Introduction

Large language models (LLMs) perform remarkably well across many domains by modeling complex interactions among features[1]. Interactions are critical for complex tasks like protein design, drug discovery, or medical diagnosis, which require examining combinations of hundreds of features. As LLMs are increasingly used in such high-stakes applications, they require trustworthy explanations to aid in responsible decision-making. Moreover, LLM explanations enable debugging and can drive development through improved understanding (Zhang et al., 2023a).

Current post-hoc explainability approaches for LLMs fall into two categories: (i) methods like Shapley values (Lundberg & Lee, 2017) and LIME (Ribeiro et al., 2016) compute *marginal* feature attribution but do not consider interactions. As a running example, consider a sentiment analysis task (see Fig. 1(a)) where the LLM classifies a review containing the sentence "Her acting never fails to impress". Marginal attribution methods miss this interaction, and instead attribute positive sentiment to "never" and "fails" (see Fig. 1(a)). (ii) *Interaction indices* such as Faith-Shap (Tsai et al., 2023) attribute interactions up to a given order $d$. That is, for $n$ input features, they compute attributions by considering all $O(n^d)$ interactions. This becomes infeasible for small $n$ and $d$. This motivates the central question of this paper:

*Can we perform interaction attribution at scale for a large input space $n$ with reasonable computational complexity?*

We answer this question affirmatively with *Spectral Explainer* (SPEX) by leveraging information-theoretic tools to efficiently identify important interactions at LLM scale. The scale of SPEX is enabled by the observation that LLM outputs are often driven by a small number of *sparse* interactions between inputs (Tsui & Aghazadeh, 2024; Ren et al., 2024a). See Fig. 1 for examples of sparsity in various tasks. SPEX discovers important interactions by using a sparse Fourier transform to construct a surrogate explanation function. This sparse Fourier transform searches for interactions via a channel decoding algorithm, thereby avoiding the exhaustive search used in existing approaches.

Our experiments show we can identify a small set of interactions that effectively and concisely reconstruct LLM outputs with $n \approx 1000$. This scale is far beyond what

---

[*]Equal contribution  [1]Department of Electrical Engineering and Computer Science, UC Berkeley [2]Department of Statistics, UC Berkeley [3]Department of Electrical and Computer Engineering, UC Santa Barbara. Correspondence to: Justin Singh Kang <justin_kang@berkeley.edu>.

*Proceedings of the 42nd International Conference on Machine Learning*, Vancouver, Canada. PMLR 267, 2025. Copyright 2025 by the author(s).

[1]LLM features refer to inputs at any granularity, e.g, tokens, sentences in a prompt or image patches in vision-language models.

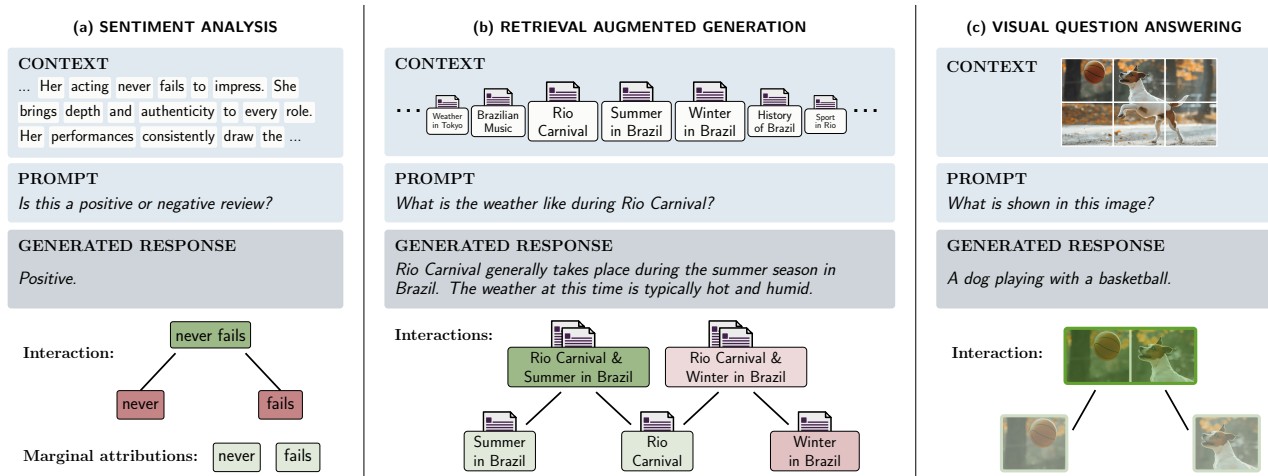

Figure 1: (a) Sentiment analysis: SPEX identifies the double negative "never fails". Marginal approaches assign positive attributions to "never" and "fails". (b) Retrieval augmented generation: SPEX explains the output of a RAG pipeline, finding a *combination* of documents the LLM used to answer the question and ignoring unimportant information. (c) Visual question answering: SPEX identifies interaction between image patches required to correctly summarize the image.

current interaction attribution benchmarks consider, e.g., SHAP-IQ (Muschalik et al., 2024) only considers datasets with no more than 20 features. This is summarized in Fig 2; marginal attribution methods scale to large $n$ but ignore crucial interactions. On the other hand, existing interaction indices do not scale with $n$. SPEX both captures interactions *and* scales to large $n$. For an $s$ sparse Fourier transform containing interactions of at most degree $d$, SPEX has computational complexity at most $\tilde{O}(sdn)$. In contrast, popular interaction attribution approaches scale as $\Omega(n^d)$.

**Evaluation Overview.** We compare SPEX to popular feature and interaction indices across three standard datasets. Algorithms and experiments are made publicly available[2].

- **Faithfulness.** SPEX more faithfully reconstructs ($\approx 20\%$ improvement) outputs of LLMs as compared to other methods across datasets. Moreover, it learns more faithful reconstructions with fewer model inferences.

- **Identifying Interactions.** SPEX identifies a small number of influential interactions that significantly change model output. For one of our datasets, *HotpotQA*, SPEX provides interactions that align with human annotations.

- **Case Studies.** We demonstrate how one might use SPEX to identify and debug reasoning errors made by closed-source LLMs (*GPT-4o mini*) and for compositional reasoning in a large multi-modal model.

---
[2]https://github.com/basics-lab/spectral-explain

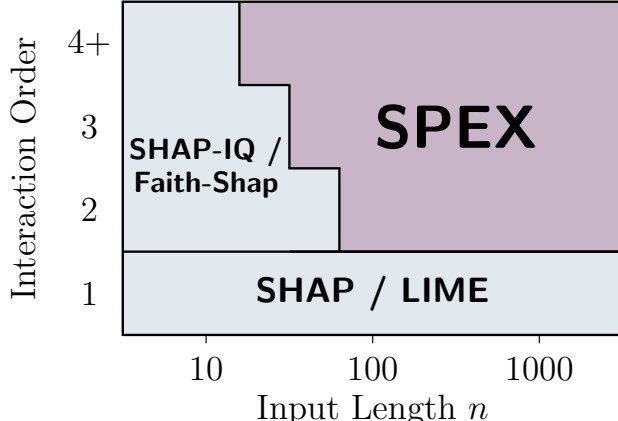

Figure 2: Marginal attribution approaches scale to large $n$, but do not capture interactions. Interaction indices only work for small $n$. SPEX computes interactions *and* scales.

## 2. Related Work

LLMs are capable of generating rationalizations for their outputs, but such rationalizations are another form of model output, susceptible to the same limitations (Sarkar, 2024). In contrast, this work focuses on explanations in the form of feature attributions that are *grounded* in the model's inputs and outputs, and can be applied to any ML model. i.e., model-agnostic methods. Moreover, model-agnostic methods can be applied to LLMs incapable of explaining their own output such as protein language models as well as encoder-only models (see experiments in Sec. 6)

**Model-Agnostic Feature Attributions** LIME (Ribeiro et al., 2016), SHAP (Lundberg & Lee, 2017), and Banzhaf

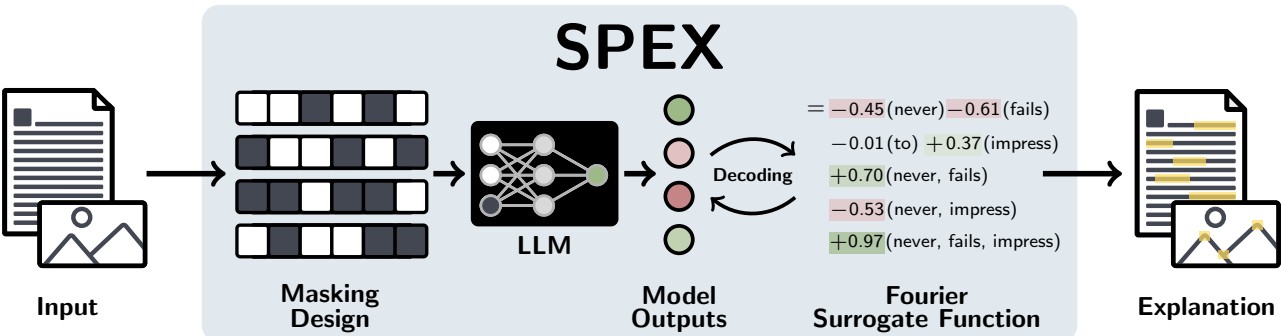

Figure 3: SPEX utilizes channel codes to determine masking patterns. We observe the changes in model output depending on the used mask. SPEX uses message passing to learn a surrogate function to generate interaction-based explanations.

values (Wang & Jia, 2023) are popular model-agnostic feature attribution approaches. SHAP and Banzhaf use game-theoretic tools for feature attribution, while LIME fits a sparse linear model. Chen et al. (2018) utilize tools from information theory for feature attributions. Other methods (Sundararajan et al., 2017; Binder et al., 2016) instead utilize internal model structure to derive feature attributions.

**Interaction Indices** Tsai et al. (2023) and Sundararajan et al. (2020) extend Shapley values to consider interactions. Fumagalli et al. (2023) provide a general framework towards interaction attribution but can only scale to at most $n \approx 20$ input features. Ren et al. (2023; 2024b) theoretically study sparse interactions, a widely observed phenomenon in practice. Kang et al. (2024) show that sparsity under the Möbius transform (Harsanyi, 1958) can be theoretically exploited for efficient interaction attribution. In practice, the proposed algorithm fails due to noise being amplified by the non-orthogonality of the Möbius basis. Our work utilizes the *orthonormal* Fourier transform, which improves robustness by preventing noise amplification. Hsu et al. (2024) apply tools from mechanistic interpretability such as circuit discovery for interaction attribution.

**Feature Attribution in LLMs** Enouen et al. (2023); Paes et al. (2024) propose hierarchical feature attribution for language models that first groups features (paragraphs) and then increase the feature space via a more fine-grained analysis (sentences or words/tokens). Cohen-Wang et al. (2024) provide marginal feature importances via LASSO. These works do not explicitly compute interaction attributions.

## 3. Overview: Fourier Transform Formulation

We review background on the Fourier transform for SPEX.

**Model Input** Let $\mathbf{x}$ be the input to the LLM where $\mathbf{x}$ consists of $n$ input features, e.g., words. For $\mathbf{x} =$ "Her acting never fails to impress", $n = 6$. In Fig. 1(b) and (c), $n$ refers to the number of documents or image patches. For $S \subseteq [n]$, we define $\mathbf{x}_S$ as a masked input where $S$ denotes

the coordinates in $\mathbf{x}$ we retain, replacing all others with the [MASK] token. For example, if $S = \{1, 2, 4, 5, 6\}$, then the masked input $\mathbf{x}_S$ is "Her acting [MASK] fails to impress". Masks can be more generally applied to any input. In Fig. 1(b) and (c), masks are applied over documents and image patches respectively.

**Value Function** For input $\mathbf{x}$, let $f(\mathbf{x}_S) \in \mathbb{R}$ be the output of the LLM under masking pattern $S$. In sentiment analysis, (see Fig. 1(a)) $f(\mathbf{x}_S)$ is the logit of the positive class. If $\mathbf{x}_S$ is "Her acting [MASK] fails to impress", this masking pattern changes the score from positive to negative. For text generation tasks, we use the well-established practice of *scalarizing* generated text using the negative log-perplexity[3] of generating the **original** output for the unmasked input $\mathbf{x}$ (Paes et al., 2024; Cohen-Wang et al., 2024). Since we only consider sample-specific explanations for a given $\mathbf{x}$, we suppress dependence on $\mathbf{x}$ and write $f(\mathbf{x}_S)$ as $f(S)$.

**Fourier Transform of Value Function** Let $\mathbb{F}_2^n = \{0, 1\}^n$, and addition between two elements in $\mathbb{F}_2$ as XOR. Since there are $2^n$ possible masks $S$, we equivalently write $f : \mathbb{F}_2^n \to \mathbb{R}$, where $f(S) = f(\mathbf{m})$ with $S = \{i : m_i = 1\}$. That is, $\mathbf{m} \in \mathbb{F}_2^n$ is a binary vector representing a *masking pattern*. If $m_i = 0$ we evaluate the model after masking the $i^{\text{th}}$ input. The Fourier transform $F : \mathbb{F}_2^n \to \mathbb{R}$ of $f$ is:

$$f(\mathbf{m}) = \sum_{\mathbf{k} \in \mathbb{F}_2^n} (-1)^{\langle \mathbf{m}, \mathbf{k} \rangle} F(\mathbf{k}). \tag{1}$$

The Fourier transform is an *orthonormal* transform onto a parity (XOR) function basis (O'Donnell, 2014).

**Sparsity** $f$ is *sparse* if $F(\mathbf{k}) \approx 0$ for most of the $\mathbf{k} \in \mathbb{F}_2^n$. Moreover, we call $f$ *low degree*, if large $F(\mathbf{k})$ have small $|\mathbf{k}|$. Ren et al. (2024b); Kang et al. (2024); Valle-Perez et al. (2018); Yang & Salman (2019) and experiments in Appendix B establish that deep-learning based value functions $f$ are sparse and low degree. See Fig. 1 for examples.

---

[3]Other approaches to scalarization exist: text embedding similarity, BERT score, etc. See (Paes et al., 2024) for an overview.

## 4. Problem Statement

Our goal is to compute an *approximate surrogate* $\hat{f}$. SPEX finds a small set of $\mathbf{k}$ with $|\mathbf{k}| \ll n$ denoted $\mathcal{K}$, and $\hat{F}(\mathbf{k})$ for each $\mathbf{k} \in \mathcal{K}$ such that

$$\hat{f}(\mathbf{m}) = \sum_{\mathbf{k} \in \mathcal{K}} (-1)^{\langle \mathbf{m}, \mathbf{k} \rangle} \hat{F}(\mathbf{k}). \qquad (2)$$

This goal is motivated by the Fourier sparsity that commonly occurs in real-world data and models. Some current interaction indices (Tsai et al., 2023) determine $\mathcal{K}$ by formulating it as a LASSO problem, and solving it via $\ell_1$-penalized regression (Tibshirani, 1996),

$$\hat{F} = \operatorname*{argmin}_{\hat{F}} \sum_{\mathbf{m}} \left| f(\mathbf{m}) - \hat{f}(\mathbf{m}) \right|^2 + \lambda \left\| \hat{F} \right\|_1. \qquad (3)$$

For given order $d$, this approach requires enumeration of all $O(n^d)$ interactions. This leads to an explosion in computational complexity as $n$ grows, as confirmed by our experiments (see Fig 4(a)). To resolve this problem, we need to find an efficient way to search the space of interactions.

**Ideas behind SPEX** The key to efficient search is realizing that we are not solving an *arbitrary* regression problem: (i) the Fourier transform (1) imparts algebraic structure and (ii) we can design the masking patterns $\mathbf{m}$ with sparsity and that structure in mind. SPEX exploits this by embedding a BCH Code (Lin & Costello, 1999), a widely used algebraic channel code, into the masking patterns. In doing so, we map the problem of searching the space of interactions onto the problem of decoding a message (the important $\mathbf{k}$) from a noisy channel. We decode via the Berlekamp-Massey algorithm (Massey, 1969), a well-established algebraic algorithm for decoding BCH codes.

## 5. SPEX: Algorithm Overview

We now provide a brief overview of SPEX (see Fig. 3). A complete overview is provided in Appendix A. The high-level description consists of three parts:

**Step 1**: Determine a minimal set of masking patterns $\mathbf{m}$ to use for model inference, and query $f(\mathbf{m})$ for each $\mathbf{m}$.

**Step 2**: Efficiently learn the surrogate function $\hat{f}$ from the set of collected samples $f(\mathbf{m})$.

**Step 3**: Use $\hat{f}$ and its transform $\hat{F}$ to identify important interactions for attribution.

### 5.1. Masking Pattern Design: Exploiting Structure

We first highlight two important properties of Fourier transform related to masking design structure.

*Aliasing (Coefficient Collapse) Property*: For $b \leq n$ and $\mathbf{M} \in \mathbb{F}_2^{b \times n}$, let $u : \mathbb{F}_2^b \to \mathbb{R}$ denote a subsampled version of

$f$. Then $u$ has Fourier transform $U$:

$$u(\boldsymbol{\ell}) = f(\mathbf{M}^\top \boldsymbol{\ell}) \iff U(\mathbf{j}) = \sum_{\mathbf{Mk}=\mathbf{j}} F(\mathbf{k}). \qquad (4)$$

*Shift Property*: For any function $f : \mathbb{F}_2^n \to \mathbb{R}$, if we shift the input by some vector $\mathbf{p} \in \mathbb{F}_2^n$, the Fourier transform changes as follows:

$$f_{\mathbf{p}}(\mathbf{m}) = f(\mathbf{m} + \mathbf{p}) \iff F_{\mathbf{p}}(\mathbf{k}) = (-1)^{\langle \mathbf{p}, \mathbf{k} \rangle} F(\mathbf{k}). \quad (5)$$

**Designing Aliasing** The aliasing property (4) dictates that when sampling according to $\mathbf{M} \in \mathbb{F}_2^{b \times n}$, all $F(\mathbf{k})$ with image $\mathbf{j} = \mathbf{Mk}$ are added together. If only *one* dominant $F(\mathbf{k})$ satisfies $\mathbf{Mk} = \mathbf{j}$, which can happen due to sparsity, we call it a *singleton*. We want $\mathbf{M}$ to maximize the number of singletons, since we ultimately use singletons to recover the dominant coefficients and estimate $\hat{F}$. SPEX uses $\mathbf{M}$ with elements chosen uniformly from $\mathbb{F}_2$. Such $\mathbf{M}$ has favorable properties regarding generating singletons.

**Designing Shifts** Once we create singletons, we need to identify them, extract the dominant index $\mathbf{k}$, and estimate $\hat{F}(\mathbf{k})$. The shift property (5) is critical for this task since the sign of the dominant $F(\mathbf{k})$ changes depending on $\langle \mathbf{p}, \mathbf{k} \rangle$. Thus, each time we apply a shift vector, we gather (potentially noisy) information about the dominant $\mathbf{k}$. Finding $\mathbf{k}$ and estimating $\hat{F}(\mathbf{k})$ can be modeled as communicating information over a noisy channel (Shannon, 1948), where the communication protocol is controlled by the shift vectors. We use the aforementioned BCH channel code, which requires only $\approx t \log(n)$ shifts to recover $\mathbf{k}$. The parameter $t$ controls the robustness of the decoding procedure. Generally, if the maximum degree is $|\mathbf{k}| = d$, we choose $t \geq d$. If $t - |\mathbf{k}| > 0$, we use the additional shifts to improve the estimation of $\hat{F}(\mathbf{k})$. Since *most* of the time $|\mathbf{k}|$ is less than 5, we fix $t = 5$ for experiments in this paper.

**Combined Masking** Combining ideas from above, we construct $C = 3$ independently sampled $\mathbf{M}_c$ and $p$ shifting vectors $\mathbf{p}_i$, which come from rows of a BCH parity matrix. Then, for $c \in [C]$ and $i \in [p]$, we entirely sample the function $u_{c,i}(\boldsymbol{\ell}) = f(\mathbf{M}_c^\top \boldsymbol{\ell} + \mathbf{p}_i)$. The total number of samples is $\approx C 2^b t \log(n)$. We note that all model inference informed by our masking pattern can be conducted in parallel. The Fourier transform of each $u_{c,i}$, denoted $U_{c,i}$, is connected to the transform of the original function via

$$U_{c,i}(\mathbf{j}) = \sum_{\mathbf{k} \,:\, \mathbf{M}_c \mathbf{k} = \mathbf{j}} (-1)^{\langle \mathbf{p}_i, \mathbf{k} \rangle} F(\mathbf{k}). \qquad (6)$$

### 5.2. Computing the Surrogate Function

Once we have the samples, we use an iterative message passing algorithm to estimate $\hat{F}(\mathbf{k})$ for a small (a-priori unknown) set of $\mathbf{k} \in \mathcal{K}$.

**Bipartite Graph** We construct a bipartite graph depicted in Fig. 5. The observations $\mathbf{U}_c(\mathbf{j}) = (U_{c,0}(\mathbf{j}), \ldots, U_{c,p}(\mathbf{j}))$ are factor nodes, while the values $\hat{F}(\mathbf{k})$ correspond to variable nodes. $\hat{F}(\mathbf{k})$ is connected to $\mathbf{U}_c(\mathbf{j})$ if $\mathbf{M}_c\mathbf{k} = \mathbf{j}$.

**Message Passing** The messages from factor to variable are computed by attempting to decode a singleton via the Berlekamp-Massey algorithm. If a $\mathbf{k}$ is successfully decoded, $\mathbf{k}$ is added to $\mathcal{K}$ and $F(\mathbf{k})$ is estimated and sent to factor node $\hat{F}(\mathbf{k})$. The variable nodes send back the average of their received messages to all connected factor nodes. The factor nodes then update their estimates of $\hat{F}$, and attempt decoding again. The process repeats until convergence. Once complete the surrogate function is constructed from $\mathcal{K}$ and $\hat{F}(\mathbf{k})$ according to (2). Complete step-by-step details are in Appendix A, Algorithm 4.

# 6. Experiments

**Datasets** We use three popular datasets that require the LLM to understand interactions between features.

1. *Sentiment* is primarily composed of the *Large Movie Review Dataset* (Maas et al., 2011), which contains both positive and negative IMDb movie reviews. The dataset is augmented with examples from the *SST* dataset (Socher et al., 2013) to ensure coverage for small $n$. We treat the words of the reviews as the input features.

2. *HotpotQA* (Yang et al., 2018) is a question-answering dataset requiring multi-hop reasoning over multiple Wikipedia articles to answer complex questions. We use the sentences of the articles as the input features.

3. *Discrete Reasoning Over Paragraphs* (DROP) (Dua et al., 2019) is a comprehension benchmark requiring discrete reasoning operations like addition, counting, and sorting over paragraph-level content to answer questions. We use the words of the paragraphs as the input features.

**Models** For *DROP* and *HotpotQA*, (generative question-answering tasks) we use `Llama-3.2-3B-Instruct` (Grattafiori et al., 2024) with 8-bit quantization. For *Sentiment* (classification), we use the encoder-only fine-tuned `DistilBERT` model (Sanh et al., 2019; Odabasi, 2025).

**Baselines** We compare against popular marginal metrics LIME, SHAP, and the Banzhaf value. For interaction indices, we consider Faith-Shapley, Faith-Banzhaf, and the Shapley-Taylor Index. We compute all benchmarks where computationally feasible. That is, we always compute marginal attributions and interaction indices when $n$ is sufficiently small. In figures, we only show the best performing baselines. Results and implementation details for all baselines can be found in Appendix B.

**Hyperparameters** SPEX has several parameters to determine the number of model inferences (masks). We choose

$C = 3$, informed by Li et al. (2014) under a simplified sparse Fourier setting. We fix $t = 5$, which is the error correction capability of SPEX and serves as an approximate bound on the maximum degree. We also set $b = 8$; the total collected samples are $\approx C2^b t \log(n)$. An empirical validation of these selections on the *sentiment* dataset is presented in Appendix B.5.1. For $\ell_1$ regression-based interaction indices, we choose the regularization parameter via 5-fold cross-validation.

## 6.1. Metrics

We compare SPEX to other methods across a variety of well-established metrics to assess performance.

**Faithfulness**: To characterize how well the surrogate function $\hat{f}$ approximates the true function, we define *faithfulness* (Zhang et al., 2023b):

$$R^2 = 1 - \frac{\|\hat{f} - f\|^2}{\|f - \bar{f}\|^2}, \qquad (7)$$

where $\|f\|^2 = \sum_{\mathbf{m}\in\mathbb{F}_2^n} f(\mathbf{m})^2$ and $\bar{f} = \frac{1}{2^n}\sum_{\mathbf{m}\in\mathbb{F}_2^n} f(\mathbf{m})$. Faithfulness measures the ability of different explanation methods to predict model output when masking random inputs. We measure faithfulness over 10,000 random *test* masks per-sample, and report average $R^2$ across samples.

**Top-$r$ Removal**: We measure the ability of methods to identify the top $r$ influential features to model output:

$$\mathrm{Rem}(r) = \frac{|f(\mathbf{1}) - f(\mathbf{m}^*)|}{|f(\mathbf{1})|},$$
$$\mathbf{m}^* = \underset{|\mathbf{m}|=n-r}{\mathrm{argmax}} |\hat{f}(\mathbf{1}) - \hat{f}(\mathbf{m})|. \qquad (8)$$

**Recovery Rate@$r$**: Each question in *HotpotQA* contains human-labeled annotations for the sentences required to correctly answer the question. We measure the ability of interaction indices to recover these human-labeled annotations. Let $S_{r^*} \subseteq [n]$ denote human-annotated sentence indices. Let $S_i$ denote feature indices of the $i^{\text{th}}$ most important interaction for a given interaction index. Define the recovery ability at $r$ for each method as follows

$$\text{Recovery@}r = \frac{1}{r}\sum_{i=1}^{r} \frac{|S_r^* \cap S_i|}{|S_i|}. \qquad (9)$$

Intuitively, (9) measures how well interaction indices capture features that align with human-labels.

## 6.2. Faithfulness and Runtime

Fig. 4 shows the faithfulness of SPEX compared to other methods. We also plot the runtime of all approaches for the *Sentiment* dataset for different values of $n$. All attribution methods are learned over a fixed number of training masks.

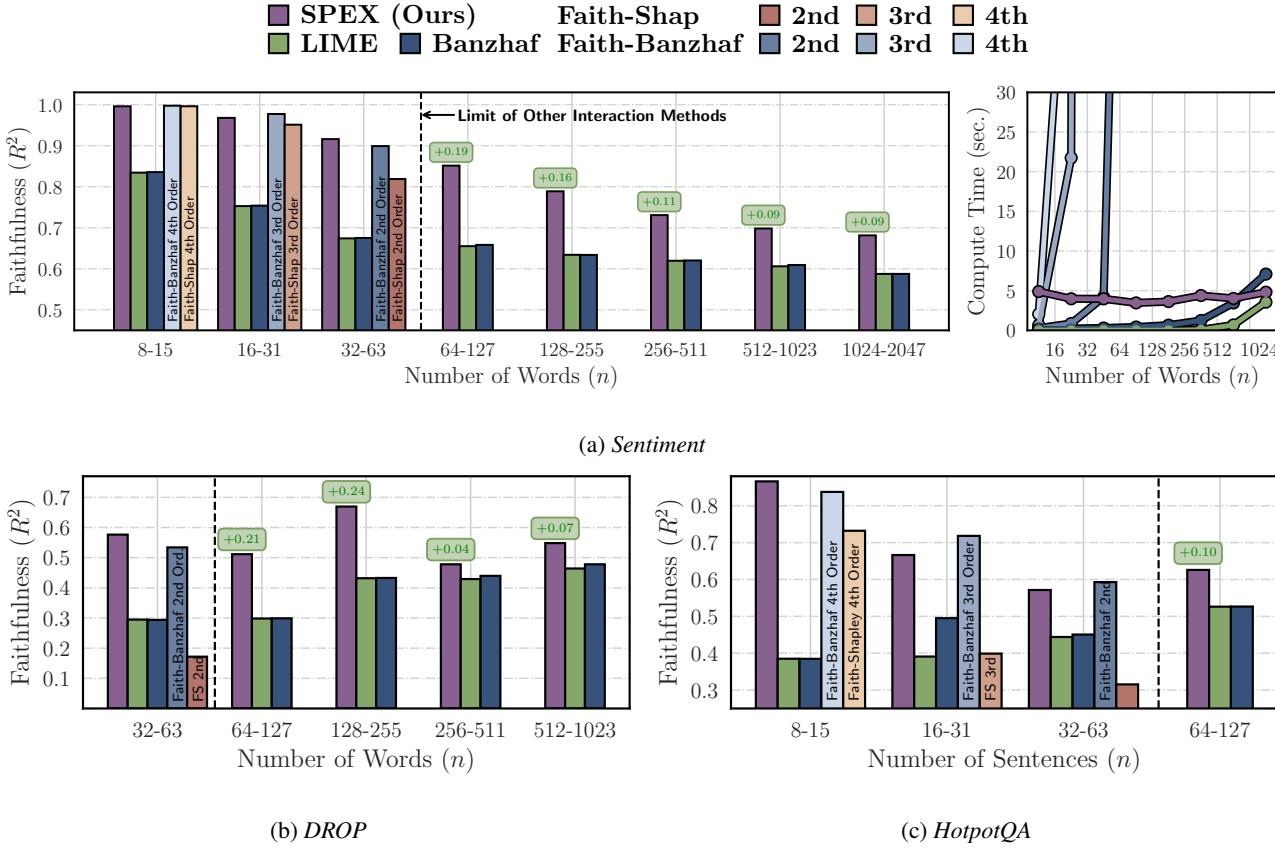

(a) *Sentiment*

(b) *DROP*

(c) *HotpotQA*

Figure 4: (a) SPEX uniformly outperforms all baselines in terms of faithfulness. High order Faith-Banzhaf indices have competitive faithfulness, but rapidly increase in computational cost. (b) The DROP dataset contains only larger examples, so we primarily compare against first order methods. (c) Our approach remains competitive in this task as well, and still outperforms marginal approaches for large $n$.

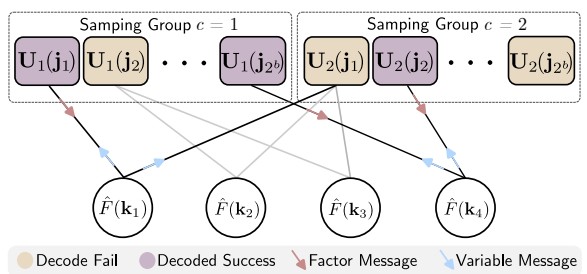

Figure 5: Depiction of the message passing algorithm for computing the surrogate function in SPEX.

**Comparison to Interaction Indices** SPEX maintains competitive performance with the best-performing interaction indices across datasets. Recall these indices enumerate *all possible interactions*, whereas SPEX does not. This difference is reflected in the runtimes of Fig. 4(a). The runtime of other interaction indices explodes as $n$ increases while SPEX does not suffer any increase in runtime.

**Comparison to Marginal Attributions** For input lengths $n$ too large to run interaction indices, SPEX is significantly more faithful than marginal attribution approaches across all three datasets.

**Varying number of training masks** Results in Appendix B show that SPEX continues to out-perform other approaches as we vary the number of training masks.

**Sparsity of SPEX Surrogate Function** Results in Appendix B, Table 3 show surrogate functions learned by SPEX have Fourier representations where only $\sim 10^{-100}$ percent of coefficients are non-zero.

### 6.3. Removal

Fig. 6 plots the change in model output as we mask the top $r$ features for different regimes of $n$.

**Small $n$** SPEX is competitive with other interaction indices for *Sentiment*, and out-performs them for *HotpotQA* and *DROP*. Performance of SPEX in this task is particularly notable since Shapley-based methods are designed to iden-

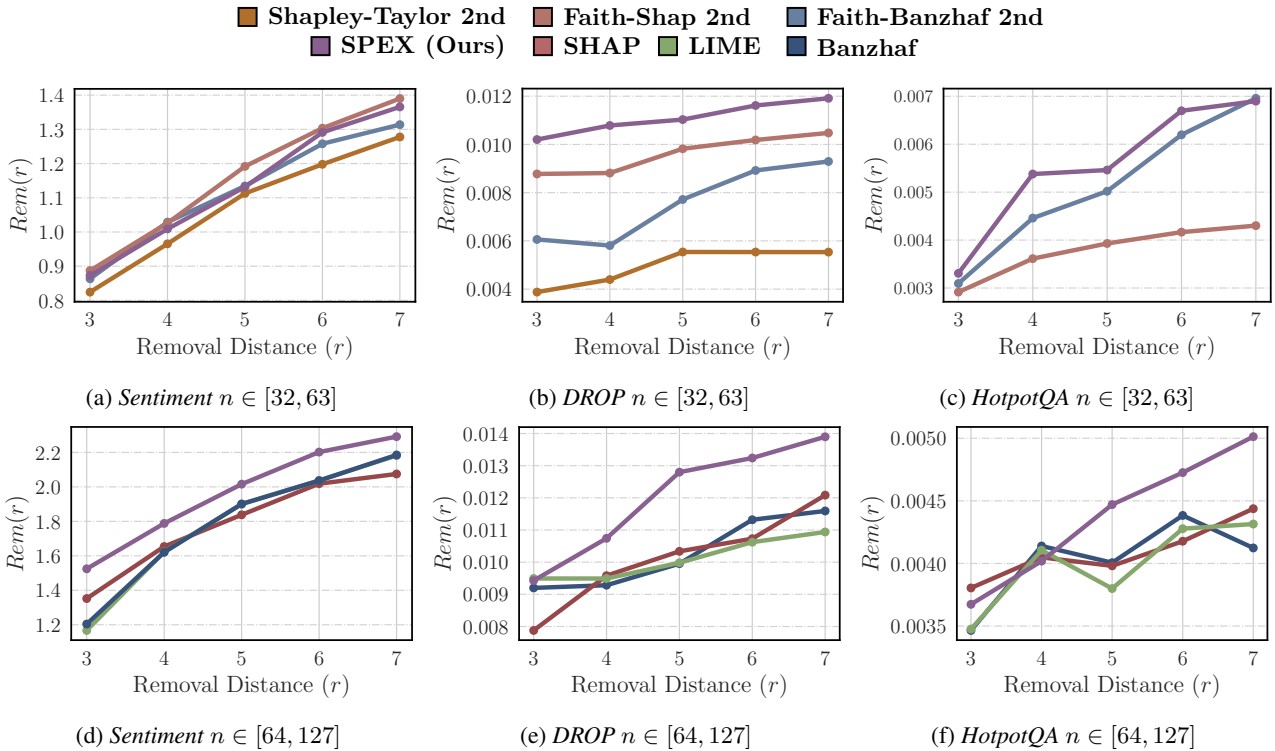

Figure 6: On the removal task, SPEX performs competitively with 2nd order methods on the *Sentiment* dataset, and out-performs all approaches on *DROP* and *HotpotQA* dataset for $n \in [32, 63]$. When $n$ is too large to compute other interaction indices, we outperform marginal methods.

tify a small set of influential features. On the other hand, SPEX does not optimize for this metric, but instead learns the function $f(\cdot)$ over all possible $2^n$ masks.

**Large $n$** SPEX out-performs all marginal approaches, indicating the utility of considering interactions.

### 6.4. Recovery Rate of Human-Labeled Interactions

We compare the recovery rate (9) for $r = 10$ of SPEX against third order Faith-Banzhaf and Faith-Shap interaction indices. We choose third order interaction indices because all examples are answerable with information from at most three sentences, i.e., maximum degree $d = 3$. Recovery rate is measured as we vary the number of training masks.

Results are shown in Fig. 7a, where SPEX has the highest recovery rate of all interaction indices across all sample sizes. Further, SPEX achieves close to its maximum performance with few samples, other approaches require many more samples to approach the recovery rate of SPEX.

**Example of Learned Interaction by SPEX** Fig. 7b displays a long-context example (128 sentences) from *HotpotQA* whose answer is contained in the three highlighted sentences. SPEX identifies the three human-labeled sentences as the most important third order interaction while ignoring unimportant contextual information. Other third order methods are not computable at this length.

## 7. Case Studies

In this section, we apply SPEX to two case studies: debugging incorrect responses and visual question answering. Refer to Appendix B for further details on implementation.

### 7.1. Debugging Incorrect LLM Responses

LLMs often struggle to correctly answer modified versions of popular puzzle questions, even when these alterations trivialize the problem (Williams & Huckle, 2024). In this spirit, we consider a variant of the classic trolley problem:

> A runaway trolley is heading **away** from five people who are tied to the track and cannot move. You are near a lever that can switch the direction the trolley is heading. Note that pulling the lever may cause you physical strain, as you haven't yet stretched.
>
> **True or False: You should not pull the lever.**

`GPT-4o mini` (OpenAI, 2024) incorrectly selects the answer false 92.1% of the time. To understand the response, we run SHAP and SPEX over a value function that measures the logit associated to the output true.

Fig. 8 presents the results of these methods: words and interactions highlighted in green contribute positively to

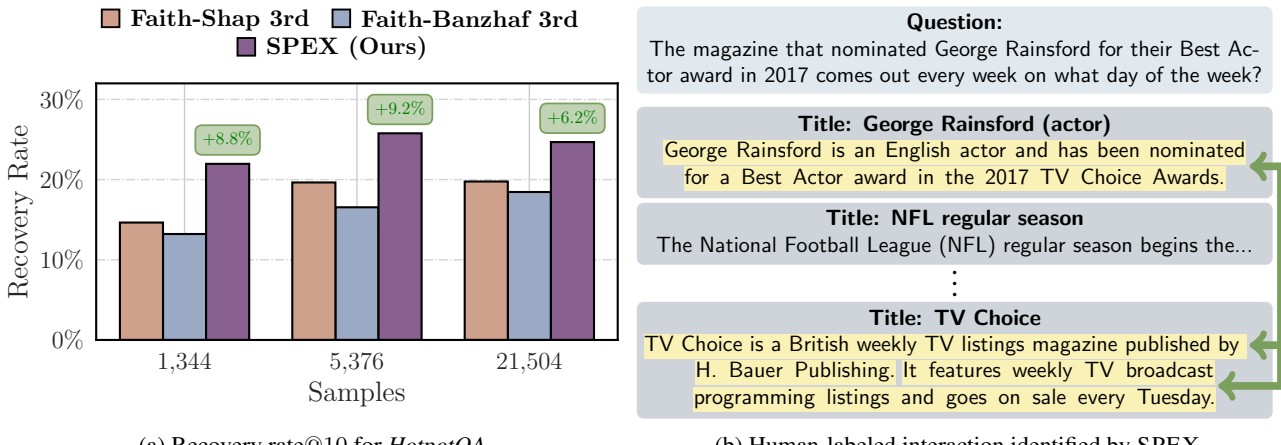

(a) Recovery rate@10 for *HotpotQA*

(b) Human-labeled interaction identified by SPEX.

Figure 7: (a) SPEX recovers more human-labeled features with significantly fewer training masks as compared to other methods. (b) For a long-context example ($n = 128$ sentences), SPEX identifies the three human-labeled sentences as the most important third order interaction while ignoring unimportant contextual information.

producing the correct output, while those in red lead the model toward an incorrect response. SHAP indicates that both instances of the word *trolley* have the most significant negative impact, while the last sentence appears to aid the model in answering correctly. A more comprehensive understanding is provided by the top interactions learned via SPEX. These interactions indicate a negative fourth order interaction involving the two instances of *trolley*, as well as the words *pulling* and *lever*. This negative interaction is emblematic of the original problem's formulation, indicating that the model may be over-fit.

### 7.2. Visual Question Answering

VQA involves answering questions based on an image. Petsiuk et al. (2018); Frank et al. (2021); Parcalabescu & Frank (2023) consider model-agnostic methods for attributing the marginal contributions of image regions to the generated response. In many compositional reasoning tasks, interactions are key and marginal attributions are insufficient. We illustrate this using an image of a dog playing with a basketball and prompting the `LLaVA-NeXT-Mistral-7B` model (Liu et al., 2023) with *"What is shown in this image?"*. This yields the response *"A dog playing with a basketball."*.

In Fig. 8, SHAP indicates that image patches containing the ball and the dog are important, but does not capture their interactions. Positive interactions obtained via SPEX reveal that the presence of both the dog and the basketball together contributes significantly more to the response than the sum of their individual contributions. This suggests that the model not only recognizes the dog and the basketball as separate objects but also understands their interaction—dog playing with the ball—as crucial for forming the correct

response. Negative interactions between different parts of the dog indicate redundancy, implying that the total effect of these regions is less than the sum of their marginal contributions.

## 8. Conclusion

Identifying feature interactions is a critical problem in machine learning. We have proposed SPEX, the first interaction based model-agnostic post-hoc explanation algorithm that is able to scale to over 1000 features. SPEX achieves this by making a powerful connection to the field of channel coding. This enables SPEX to avoid the $O(n^d)$ complexity that existing feature interaction attribution algorithms suffer from. Our experiments show SPEX is able to significantly outperform other methods across the *Sentiment*, *Drop* and *HotpotQA* datasets in terms of faithfulness, feature removal, and interaction recovery rate.

**Limitations**   Sparsity is central to our algorithm, and without an underlying sparse structure, SPEX can fail. Furthermore, even though we make strides in terms of sample efficiency, the number of samples still might remain too high for many applications, particularly when inference cost or time is high. Another consideration is the degree of human understanding we can extract from computed interactions. Manually parsing interactions can be slow, and useful visualizations of interactions vary by modality. Further improvements in visualization and post-processing of interactions are needed.

**Future Work**   SPEX works in a non-adaptive fashion, predetermining the masking patterns **m**. For greater sample efficiency, *adaptive* algorithms might be considered, where

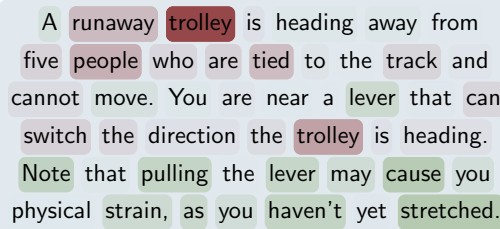 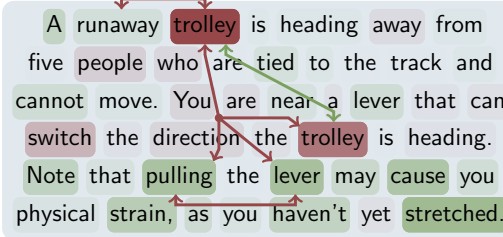

**SHAP**                    **Interactions via SPEX**

**True** or **False:** You should not pull the lever.

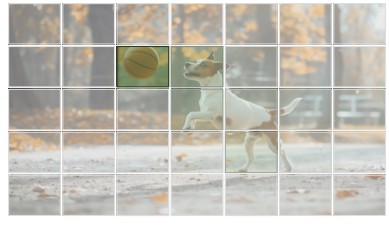 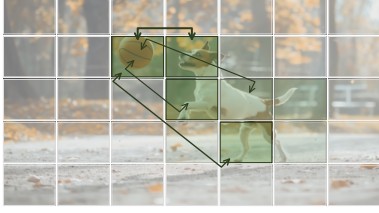 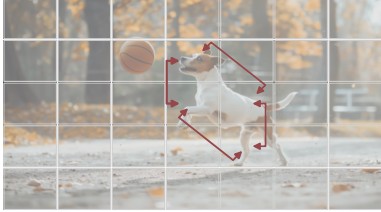

**SHAP**          **Positive Interactions via SPEX**          **Negative Interactions via SPEX**

**Query:** What is shown in this image?    **Response:** A dog playing with a basketball.

Figure 8: SHAP provides marginal feature attributions. Feature interaction attributions computed by SPEX provide a more comprehensive understanding of (above) words interactions that cause the model to answer incorrectly and (below) interactions between image patches that informed the model's output.

initial model inferences help determine future masking patterns. In addition, we have focused on *model-agnostic* explanations, but future work could consider combining this with internal model structure. Finally, interactions are a central aspect of the *attention* structures in transformers. Studying the connection between SPEX and sparse attention (Chen et al., 2021) is another direction for future research.

## Impact Statement

Getting insights into the decisions of deep learning models offers significant advantages, including increased trust in model outputs. By reasoning about the rationale behind a model's decisions with the help of SPEX, we can develop greater confidence in its output, and use it to aid in our own reasoning. When using analysis tools like SPEX, it's crucial to avoid over-interpretation of results.

## Acknowledgments

R.P. acknowledges the support of NSF award 2342253 and 2419982. K.R. acknowledges the support of NSF award 2232146. B.Y. acknowledges partial support from NSF grant DMS-2413265, NSF grant DMS 2209975, NSF grant 2023505 on Collaborative Research: Foundations of Data Science Institute (FODSI), the NSF and the Simons Foundation for the Collaboration on the Theoretical Foundations of Deep Learning through awards DMS-2031883 and 814639, NSF grant MC2378 to the Institute for Artificial CyberThreat Intelligence and OperatioN (ACTION), and NIH grant R01GM152718. L.B. acknowledges that this material is based upon work supported by the National Science Foundation Graduate Research Fellowship Program (NSF GRFP) under Grant No. DGE-2146752.

Any opinions, findings, conclusions, or recommendations expressed in this material are those of the author(s) and do not necessarily reflect the views of the National Science Foundation.

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

# A. Algorithm Details

## A.1. Introduction

This section provides the algorithmic details behind SPEX. The algorithm is derived from the sparse Fourier (Hadamard) transformation described in Li et al. (2014). Many modifications have been made to improve the algorithm and make it suitable for use in this application. Application of the original algorithm proposed in Li et al. (2014) fails for all problems we consider in this paper. In this work, we focus on applications of SPEX and defer theoretical analysis to future work.

**Relevant Literature on Sparse Transforms**  This work develops the literature on sparse Fourier transforms. The first of such works are (Hassanieh et al., 2012; Stobbe & Krause, 2012; Pawar & Ramchandran, 2013). The most relevant literature is that of the sparse Boolean Fourier (Hadamard) transform (Li et al., 2014; Amrollahi et al., 2019). Despite the promise of many of these algorithms, their application has remained relatively limited, being used in only a handful of prior applications. Our code base is forked from that of (Erginbas et al., 2023). In this work we introduce a series of major optimizations which specifically target properties of explanation functions. By doing so, our algorithm is made significantly more practical and robust than any prior work.

**Importance of the Fourier Transform**  The Fourier transform does more than just impart critical algebraic structure. The orthonormality of the Fourier transform means that small noisy variations in $f$ remain small in the Fourier domain. In contrast, AND interactions, which operate under the non-orthogonal Möbius transform (Kang et al., 2024), can amplify small noisy variations, which limits practicality. Fortunately, this is not problematic, as it is straightforward to generate AND interactions from the surrogate function $\hat{f}$. Many popular interaction indices have simple definitions in terms of $F$. Table 1 highlights some key relationships, and Appendix C provides a comprehensive list.

| Shapley Value | Banzhaf Interaction Index | Möbius Coefficient |
|---|---|---|
| $\text{SV}(i) = \sum\limits_{S \ni i,\ \|S\| \text{ odd}} F(S)/\|S\|$ | $I^{BII}(S) = (-2)^{\|S\|} F(S)$ | $I^M(S) = (-2)^{\|S\|} \sum\limits_{T \supseteq S} F(T)$ |

Table 1: Popular attribution scores in terms of Fourier coefficients

## A.2. Directly Solving the LASSO

Before we proceed, we remark that in cases where $n$ is not too large, and we expect the degree of nonzero $|\mathbf{k}| \le d$ to be reasonably small, enumeration is actually not infeasible. In such cases, we can set up the LASSO problem directly:

$$\hat{F} = \underset{\tilde{F}}{\arg\min} \sum_{\mathbf{m}} \left| f(\mathbf{m}) - \sum_{|\mathbf{k}| \le d} \tilde{F}(\mathbf{k}) \right|^2 + \lambda \left\| \tilde{F} \right\|_1. \tag{10}$$

Note that this is distinct from the *Faith-Banzhaf* and *Faith-Shapley* solution methods because those perform regression over the AND, Möbius basis. We observe that the formulation above typically outperforms these other approaches in terms of faithfulness, likely due to the properties of the Fourier transform.

Most popular solvers use *coordinate descent* to solve (10), but there is a long line of research towards efficiently solving this problem. In our code, we also include an implementation of Approximate Message Passing (AMP) (Maleki, 2010), which can be much faster in many cases. Much like the final phase of SPEX, AMP is a low complexity message passing algorithm where messages are iteratively passed between factor nodes (observations) and variable nodes.

A more refined version of SPEX, would likely examine the parameters $n$ and the maximum degree $d$ and determine whether or not to directly solve the LASSO, or to apply the full SPEX, as we describe in the following sections.

## A.3. Masking Pattern Design and Model Inference

The first part of the algorithm is to determine which samples we collect. All steps of this part of the algorithm are outlined in Algorithm 1. This is governed by two structures: the random linear codes $\mathbf{M}_c$ and the BCH parity matrix $\mathbf{P}$. Random linear codes have been well studied as central objects in error correction and cryptography. They have previously been considered

for sparse transforms in (Amrollahi et al., 2019). They are suitable for this application because they roughly uniformly hash $\mathbf{k}$ with low hamming weight.

The use of the $\mathbf{P} \in \mathbb{F}_2^{p \times n}$, the parity matrix of a binary BCH code is novel. These codes are well studied for the applications in error correction (Lin & Costello, 1999), and they were once the preeminent form of error correction in digital communications. A primitive, narrow-sense BCH code is characterized by its length, denoted $n_c$, dimension, denoted $k_c$ (which we want to be equal to our input dimension $n$) and its error correcting capability $t_c = 2d + 1$, where $d$ is the minimum distance of the code. For some integer $m > 3$ and $t_c < 2^{m-1}$, the parameters satisfy the following equations:

$$n_c = 2^m - 1 \tag{11}$$

$$p = n_c - k_c \leq mt. \tag{12}$$

Note that the above says we can bounds $p \leq t \lceil \log_2(n_c) \rceil$, and it is easy to solve for $p$ given $n = k_c$ and $t$, however, explicitly bounding $p$ in terms of $n$ and $t$ is difficult, so for the purpose of discussion, we simply write $p \approx t \log(n)$, since $n_c = p + n$, and we expect $n \gg p$ in nearly all cases of interest.

We use the software package `galois` (Hostetter, 2020) to construct a generator matrix, $\mathbf{G} \in \mathbb{F}_2^{n_c \times k_c}$ in systematic form:

$$\mathbf{G} = \begin{bmatrix} \mathbf{I}_{k_c \times k_c} \\ \mathbf{P} \end{bmatrix} \tag{13}$$

Note that according to (13) $\mathbf{P} \in \mathbb{F}_2^{p \times k_c}$. In cases where $k_c > n$, we consider only the first $n$ rows of $\mathbf{P}$. This is a process

---

**Algorithm 1** Collect Samples

1: **Input:** Parameters $(n, t, b, C = 3, \gamma = 0.9)$, Query function $f(\cdot)$
2: **for** $j = 1$ **to** $n$, $i = 1$ **to** $b$, $c = 1$ **to** $C$ **do**                    ▷ Generate random linear code
3:     $X_{ij} \sim \text{Bern}(0.5)$
4:     $[\mathbf{M}_c]_{i,j} \leftarrow X_{i,j}$
5: **end for**
6: Code $\leftarrow \text{BCH}(n_c = n_c, k_c \geq n, t_c = t)$         ▷ Systematic BCH code with dimension $n$ and correcting capacity $t$
7: $p \leftarrow n_c - n$
8: $\mathbf{P} \leftarrow \text{Code}.\mathbf{P}$
9: $\mathcal{P} \leftarrow \text{rows}(\mathbf{P}) = [\mathbf{0}, \mathbf{p}_1, \ldots, \mathbf{p}_p]$
10: **for all** $\boldsymbol{\ell} \in \mathbb{F}_2^b$, $i \in \{0, \ldots, p\}$, $c \in \{1, \ldots, C\}$ **do**
11:     $u_{c,i}(\boldsymbol{\ell}) \leftarrow f\left(\mathbf{M}_c^\top \boldsymbol{\ell} + \mathcal{P}[i]\right)$                    ▷ Query the model at masking patterns
12: **end for**
13: **for all** $i \in \{0, \ldots, p\}$, $c \in \{1, \ldots, C\}$ **do**
14:     $U_{c,i} \leftarrow \text{FFT}(u_{c,i})$              ▷ Compute the Boolean Fourier transform of the collected samples
15: **end for**
16: $\mathbf{U}_c \leftarrow [U_{c,1}, \ldots, U_{c,p}]$
17: **Output:** Processed Samples $\mathbf{U}_c, U_{c,0} \ c = 1, \ldots, C$

---

known as *shortening*. Our application of this BCH code in our application is rather unique. Instead of the typical use of a BCH code as *channel correction* code, we use it as a *joint source channel code*.

Let $\mathbf{p}_0 = \mathbf{0}$, and let $\mathbf{p}_i$, $i = 1, \ldots, p$ correspond to the rows of $\mathbf{P}$. We collect samples written as:

$$u_{c,i}(\boldsymbol{\ell}) \leftarrow f\left(\mathbf{M}_c^\top \boldsymbol{\ell} + \mathbf{p}_i\right) \ \forall \boldsymbol{\ell} \in \mathbb{F}_2^b, \ c = 1, \ldots, C, \ i = 0, \ldots, p. \tag{14}$$

Note that the total number of unique samples can be upper bounded by $C(p+1)2^b$. For large $n$ this upper bound is nearly always very close to the true number of unique samples collected. After collecting each sample, we compute the boolean Fourier transform. The forward and inverse transforms as we consider in this work are defined below.

$$\text{Forward:} \quad F(\mathbf{k}) = \frac{1}{2^n} \sum_{\mathbf{m} \in \mathbb{F}_2^n} (-1)^{\langle \mathbf{k}, \mathbf{m} \rangle} f(\mathbf{m}) \qquad \text{Inverse:} \quad f(\mathbf{m}) = \sum_{\mathbf{k} \in \mathbb{F}_2^n} (-1)^{\langle \mathbf{m}, \mathbf{k} \rangle} F(\mathbf{k}), \tag{15}$$

When samples are collected according to (14), after applying the transform in (15), the transform of $u_{c,i}$ can be written as:

$$U_{c,i}(\mathbf{j}) = \sum_{\mathbf{k} \,:\, \mathbf{M}_c\mathbf{k}=\mathbf{j}} (-1)^{\langle \mathbf{p}_i, \mathbf{k} \rangle} F(\mathbf{k}). \tag{16}$$

To ease notation, we write $\mathbf{U}_c = [U_{c,1}, \ldots, U_{c,p}]^T$. Then we can write

$$\mathbf{U}_c(\mathbf{j}) = \sum_{\mathbf{k} \,:\, \mathbf{M}_c\mathbf{k}=\mathbf{j}} (-1)^{\mathbf{Pk}} F(\mathbf{k}), \tag{17}$$

where we have used the notation $(-1)^{\mathbf{Pk}} = [(-1)^{\langle \mathbf{p}_0, \mathbf{k} \rangle}, \ldots, (-1)^{\langle \mathbf{p}_p, \mathbf{k} \rangle}]^T$. We call the $(-1)^{\mathbf{Pk}}$ the *signature* of $\mathbf{k}$. This signature helps to identify the index of the largest interactions $\mathbf{k}$, and is central to the next part of the algorithm. Note that we also keep track of $U_{c,0}(\mathbf{j})$, which is equal to the unmodulated sum $U_{c,0}(\mathbf{j}) = \sum_{\mathbf{k} \,:\, \mathbf{M}_c\mathbf{k}=\mathbf{j}} F(\mathbf{k})$.

### A.4. Message Passing for Fourier Transform Recovery

Using the samples (17), we aim to recover the largest Fourier coefficients $F(\mathbf{k})$. To recover these samples we apply a message passing algorithm, described in detail in Algorithm 4. The factor nodes are comprised of the $C2^b$ vectors $\mathbf{U}_c(\mathbf{j}) \; \forall \mathbf{j} \in \mathbb{F}_2^b$. Each of these factor nodes are connected to all values $\mathbf{k}$ that are comprise their sum, i.e., $\{\mathbf{k} \mid \mathbf{M}_c\mathbf{k} = \mathbf{j}\}$. Since the number of variable nodes is too great, we initialize the value of each variable node, which we call $\hat{F}(\mathbf{k})$ to zero implicitly. The values $\hat{F}(\mathbf{k})$ for each variable node indexed by $\mathbf{k}$ represent our estimate of the Fourier coefficients.

#### A.4.1. THE MESSAGE FROM FACTOR TO VARIABLE

Consider an arbitrary factor node $\mathbf{U}_c(\mathbf{j})$ initialized according to (17). We want to understand if there are any large terms $F(\mathbf{k})$ involved in the sum in (17). To do this, we can utilize the signature sequences $(-1)^{\mathbf{Pk}}$. If $\mathbf{U}_c(\mathbf{j})$ is strongly correlated with the signature sequence of a given $\mathbf{k}$, i.e., if $\left| \langle (-1)^{\mathbf{Pk}}, \mathbf{U}_c(\mathbf{j}) \rangle \right|$ is large, and $\mathbf{M}_c\mathbf{k} = \mathbf{j}$, from the perspective of $\mathbf{U}_c(\mathbf{j})$, it is likely that $F(\mathbf{k})$ is *large*. Searching through all $\mathbf{M}_c\mathbf{k} = \mathbf{j}$, which, for a full rank $\mathbf{M}_c$ contains $2^{n-b}$ different $\mathbf{k}$ is intractable, and likely to identify many spurious correlations. Instead, we rely on the structure of the BCH code from which $\mathbf{P}$ is derived to solve this problem.

**BCH Hard Decoding**  The BCH decoding procedure is based on an idea known generally in signal processing as "treating interference as noise". For the purpose of explanation, assume that there is some $\mathbf{k}^*$ with large $F(\mathbf{k}^*)$, and all other $\mathbf{k}$ such that $\mathbf{M}_c\mathbf{k} = \mathbf{j}$ correspond to small $F(\mathbf{k})$. For brevity let $\mathcal{A}_c(\mathbf{j}) = \{\mathbf{k} \mid \mathbf{M}_c\mathbf{k} = \mathbf{j}\}$. We can write:

$$\mathbf{U}_c(\mathbf{j}) = F(\mathbf{k}^*)(-1)^{\mathbf{Pk}^*} + \sum_{\mathcal{A}_c(\mathbf{j}) \backslash \mathbf{k}^*} (-1)^{\mathbf{Pk}} F(\mathbf{k}) \tag{18}$$

After we normalize with respect to $U_{c,0}(\mathbf{j})$ this yields:

$$\frac{\mathbf{U}_c(\mathbf{j})}{U_{c,0}(\mathbf{j})} = \left( \frac{1}{1 + \sum_{\mathcal{A}_c(\mathbf{j}) \backslash \mathbf{k}^*} F(\mathbf{k})/F(\mathbf{k}^*)} \right) (-1)^{\mathbf{Pk}^*} + \left( \frac{\sum_{\mathcal{A}_c(\mathbf{j}) \backslash \mathbf{k}^*} (-1)^{\mathbf{Pk}} F(\mathbf{k})}{F(\mathbf{k}^*) + \sum_{\mathcal{A}_c(\mathbf{j}) \backslash \mathbf{k}^*} F(\mathbf{k})} \right) \tag{19}$$

$$= A(\mathbf{j})(-1)^{\mathbf{Pk}^*} + \mathbf{w}(\mathbf{j}). \tag{20}$$

As we can see, the ratio (20) is a noise-corrupted version of the signature sequence of $\mathbf{k}^*$. To estimate $\mathbf{Pk}$ we apply a nearest-neighbor estimation rule outlined in Algorithm 2. In words, if the $i$th coordinate of the vector (20) is closer to $-1$ we estimate that the corresponding element of $\mathbf{Pk}$ to be 1, conversely, if the $i$th coordinate is closer to 1 we estimate the corresponding entry to be 0. This process effectively converts the multiplicative noise $A$ and additive noise $\mathbf{w}$ to a noise vector in $\mathbb{F}_2$. We can write this as $\mathbf{Pk}^* + \mathbf{n}$. According to the Lemma A.1 if the hamming weight $\mathbf{n}$ is not too large, we can recover $\mathbf{k}^*$.

**Lemma A.1.** *If* $|\mathbf{n}| + |\mathbf{k}^*| \leq t$, *where* $\mathbf{n}$ *is the additive noise in* $\mathbb{F}_2$ *induced by the noisy process in* (20) *and the estimation procedure in Algorithm 2, then we can recover* $\mathbf{k}^*$.

*Proof.* Observe that the generator matrix of the BCH code is given by (13). Thus, there exists a codeword of the form

$$\mathbf{c} = \mathbf{Gk}^* = \begin{bmatrix} \mathbf{k}^* \\ \mathbf{Pk}^* \end{bmatrix} \tag{21}$$

Now construct the "received codeword" as in Algorithm 2:

$$\mathbf{r} = \begin{bmatrix} \mathbf{0} \\ \mathbf{P}\mathbf{k}^* + \mathbf{n} \end{bmatrix} \tag{22}$$

Thus $|\mathbf{c} - \mathbf{r}| = |\mathbf{n}| + |\mathbf{k}^*|$. Since the BCH code was designed to be $t$ error correcting, Decoding the code will recover $\mathbf{c}$, which contains $\mathbf{k}^*$. □

For decoding we use the implementation in the python package `galois` (Hostetter, 2020). It implements the standard procedure of the Berlekamp-Massey Algorithm followed by the Chien Search algorithm for BCH decoding.

---

**Algorithm 2** BCH Hard Decode

---

1: **Input:** Observation $\mathbf{U}_c(\mathbf{j})$, Decoding function $\mathrm{Dec}(\cdot)$
2: $r_i \leftarrow 0 \; i = 1 \ldots, n$
3: **for all** $i \in n+1, \ldots, n+p$ **do**
4:      $r_i \leftarrow \mathbb{1}\left\{ \frac{U_{c,i}(\mathbf{j})}{U_{c,0}(\mathbf{j})} < 0 \right\}$
5: **end for**
6: dec, $\hat{\mathbf{k}} \leftarrow \mathrm{Dec}(\mathbf{r})$
7: **Output:** dec, $\hat{\mathbf{k}}$

---

**BCH Soft Decoding** In practice the conversion of the real-valued noisy observations (20) to noisy elements in $\mathbb{F}_2$ is a process that destroys valuable information. In coding theory, this is known as *hard input* decoding, which is typically suboptimal. For example, certain coordinates will have values $\frac{U_{c,i}(\mathbf{j})}{U_{c,0}(\mathbf{j})} \approx 0$. For such coordinates, we have low confidence about the corresponding value of $(-1)^{\langle \mathbf{p}_i, \mathbf{k}^* \rangle}$, since it is equally close to $+1$ and $-1$. This uncertainty information is lost in the process of producing a hard input. With this so-called *soft information* it is possible to recover $\mathbf{k}^*$ even in cases where there are more than $t$ errors in the hard decoding case. We use a simple soft decoding algorithm for BCH decoding known as a chase decoder. The main idea behind a chase decoder is to perform hard decoding on the $d_{\mathrm{chase}}$ most likely hard inputs, and return the decoder output of the most likely hard input that successfully decoded. In practical setting like the ones we consider in this work, we don't have an understanding of the noise in (20). A practical heuristic is to simply look at the *margin* of estimation. In other words, if $\left| \frac{U_{c,i}(\mathbf{j})}{U_{c,0}(\mathbf{j})} \right|$ is large, we assume it has high confidence, while if it is small, we assume the confidence is low. Interestingly, if we assume $A(\mathbf{j}) = 1$ and $\mathbf{w}(\mathbf{j}) \sim \mathcal{N}(0, \sigma^2)$ in (20), then the ratio corresponds exactly to the logarithm of the likelihood ratio (LLR) $\log\left( \frac{\Pr(\langle \mathbf{p}_i, \mathbf{k}^* \rangle = 0)}{\Pr(\langle \mathbf{p}_i, \mathbf{k}^* \rangle = 1)} \right)$. For the purposes of soft decoding we interpret these ratios as LLRs. Pseudocode can be found in Algorithm 3.

*Remark: BCH soft decoding is a well-studied topic with a vast literature. Though we put significant effort into building a strong implementation of* SPEX*, we have used the simple Chase Decoder (described in Algorithm 3 below) as a soft decoder. The computational complexity of Chase Decoding scales as $2^{d_{chase}}$, but other methods exist with much lower computational complexity and comparable performance.*

If we successfully decode some $\mathbf{k}$ from the BCH decoding process via the bin $\mathbf{U}_c(\mathbf{j})$, we construct a message to the corresponding variable node. Before we do this, we verify that the $\mathbf{k}$ term satisfies $\mathbf{M}_c \mathbf{k} = \mathbf{j}$. This acts as a final check to increase our confidence in the output of $\mathbf{k}$. The message we construct is of the following form:

$$\mu_{(c,\mathbf{j}) \to \mathbf{k}} = \langle (-1)^{\mathbf{P}\mathbf{k}}, \mathbf{U}_c(\mathbf{j}) \rangle / p \tag{23}$$

To understand the structure of this message. This message can be seen as an estimate of the Fourier coefficient. Let's assume we are computing this message for some $\mathbf{k}^*$:

$$\mu_{(c,\mathbf{j}) \to \mathbf{k}^*} = F(\mathbf{k}^*) + \sum_{\mathcal{A}(\mathbf{j}) \backslash \mathbf{k}^*} \underbrace{\frac{1}{p} \langle (-1)^{\mathbf{P}\mathbf{k}}, (-1)^{\mathbf{P}\mathbf{k}^*} \rangle F(\mathbf{k})}_{\text{typically small}} \tag{24}$$

The inner product serves to reduce the noise from the other coefficients in the sum.

---

**Algorithm 3** BCH Soft Decode (Chase Decoding)

---

1: **Input:** Observation $\mathbf{U}_c(\mathbf{j})$, Decoding function $\mathrm{Dec}(\cdot)$, Chase depth $d_{\mathrm{chase}}$.
2: $r_i \leftarrow 0 \ i = 1 \ldots, n$
3: $\mathcal{R} \leftarrow d_{\mathrm{chase}}$ most likely hard inputs       $\triangleright$ Can be computed efficiently via dynamic programming
4: $\mathrm{dec} \leftarrow False$
5: $j \leftarrow 0$
6: **while** $dec$ is $False$ and $j \leq d_{\mathrm{chase}}$ **do**
7:    $\mathbf{r}_{(n+1):(n+p)} \leftarrow \mathcal{R}[j]$
8:    $j \leftarrow j + 1$
9:    $\mathrm{dec}, \hat{\mathbf{k}} \leftarrow \mathrm{Dec}(\mathbf{r})$
10: **end while**
11: **Output:** $\mathrm{dec}, \hat{\mathbf{k}}$

---

---

**Algorithm 4** Message Passing

---

1: **Input:** Processed Samples $\mathbf{U}_c, c = 1, \ldots, C$
2: $\mathcal{S} = \left\{ (c, \mathbf{j}) : \mathbf{j} \in \mathbb{F}_2^b, c \in \{1, \ldots, C\} \right\}$          $\triangleright$ Nodes to process
3: $\hat{F}[\mathbf{k}] \leftarrow 0 \ \forall \mathbf{k}$
4: $\mathcal{K} \leftarrow \emptyset$
5: **while** $|\mathcal{S}| > 0$ **do**                  $\triangleright$ Outer Message Passing Loop
6:    $\mathcal{S}_{\mathrm{sub}} \leftarrow \emptyset$
7:    $\mathcal{K}_{\mathrm{sub}} \leftarrow \emptyset$
8:    **for** $(c, \mathbf{j}) \in \mathcal{S}$ **do**
9:      $\mathrm{dec}, \mathbf{k} \leftarrow \mathrm{DecBCH}(\mathbf{U}_c(\mathbf{j}))$          $\triangleright$ Process Factor Node
10:      **if** $\mathrm{dec}$ **then**
11:        $\mathrm{corr} \leftarrow \frac{\langle (-1)^{\mathbf{P}\mathbf{k}}, \mathbf{U}_c(\mathbf{j}) \rangle}{\|\mathbf{U}_c(\mathbf{j})\|^2}$
12:      **else**
13:        $\mathrm{corr} \leftarrow 0$
14:      **end if**
15:      **if** $\mathrm{corr} > \gamma$ **then**             $\triangleright$ Interaction identified
16:        $\mathcal{S}_{\mathrm{sub}} \leftarrow \mathcal{S}_{\mathrm{sub}} \cup \{(\mathbf{k}, c, \mathbf{j})\}$
17:        $\mathcal{K}_{\mathrm{sub}} \leftarrow \mathcal{K}_{\mathrm{sub}} \cup \{\mathbf{k}\}$
18:      **else**
19:        $\mathcal{S} \leftarrow \mathcal{S} \setminus \{(c, \mathbf{j})\}$           $\triangleright$ Cannot extract interaction
20:      **end if**
21:    **end for**
22:    **for** $\mathbf{k} \in \mathcal{K}_{\mathrm{sub}}$ **do**
23:      $\mathcal{S}_{\mathbf{k}} \leftarrow \{(\mathbf{k}', c', \mathbf{j}') \mid (\mathbf{k}', c', \mathbf{j}') \in \mathcal{S}_{\mathrm{sub}}, \mathbf{k}' = \mathbf{k}\}$
24:      $\mu_{(c,\mathbf{j}) \to \mathbf{k}} \leftarrow \langle (-1)^{\mathbf{P}\mathbf{k}}, \mathbf{U}_c(\mathbf{j}) \rangle / p$
25:      $\mu_{\mathbf{k} \to \mathrm{all}} \leftarrow \sum_{(\mathbf{k}, c, \mathbf{j}) \in \mathcal{S}_{\mathbf{k}}} \mu_{(c, \mathbf{j}) \to \mathbf{k}}$
26:      $\hat{F}(\mathbf{k}) \leftarrow \hat{F}(\mathbf{k}) + \mu_{\mathbf{k} \to \mathrm{all}}$          $\triangleright$ Update variable node
27:      **for** $c \in \{1, \ldots, C\}$ **do**
28:        $\mathbf{U}_c(\mathbf{M}_c\mathbf{k}) \leftarrow \mathbf{U}_c(\mathbf{M}_c\mathbf{k}) - \mu_{\mathbf{k} \to \mathrm{all}} \cdot (-1)^{\mathbf{P}\mathbf{k}}$    $\triangleright$ Update factor node
29:        $\mathcal{S} \leftarrow \mathcal{S} \cup \{(c, \mathbf{M}_c\mathbf{k})\}$
30:      **end for**
31:    **end for**
32:    $\mathcal{K} \leftarrow \mathcal{K} \cup \mathcal{K}_{\mathrm{sub}}$
33: **end while**
34: **Output:** $\left\{ \left( \mathbf{k}, \hat{F}(\mathbf{k}) \right) \mid \mathbf{k} \in \mathcal{K} \right\}$, interactions, and scalar values corresponding to interactions.

---

A.4.2. THE MESSAGE FROM VARIABLE TO FACTOR

The message from factor to variable is comparatively simple. The variable node takes the average of all the messages it receives, adding the result to its state, and then sends that average back to all connected factor nodes. These factor nodes then subtract this value from their state and then the process repeats.

### A.5. Computational Complexity

**Generating masking patterns m** Constructing each masking pattern requires $n2^b$ for each $\mathbf{M}_c$. The algorithm for computing it efficiently involves a gray iteratively adding to an $n$ bit vector and keeping track of the output in a Gray code. Doing this for all $C$, and then adding all $p$ additional shifting vectors makes the cost $O(Cpn2^b)$.

**Taking FFT** For each $u_{c,i}$ we take the Fast Fourier transform in $b2^b$ time, with a total of $O(Cpb2^b)$. This is dominated by the previous complexity since, $b \leq n$

**Message passing** One round of BCH hard decoding is $O(n_c t + t^2)$. For soft decoding, this cost is multiplied by $2^{d_{\text{chase}}}$, which we is a constant. Computing the correlation vector is $O(np)$, dominated by the computation of $\mathbf{Pk}$. In the worst case, we must do this for all $C2^b$ vectors $\mathbf{U}_c(\mathbf{j})$. We also check that $\mathbf{Mk} = \mathbf{j}$ before sending the message, which costs $O(nb)$. Thus, processing all the factor nodes costs $O(C2^b(n_c t + t^2 + n(p + b)))$. The number of active (with messages to send) variable nodes is at most $C2^b$, and computing their factors is at most $C$. Thus, computing factor messages is at most $C^2 2^b$ messages. Finally, factor nodes are updated with at most $C2^b$ variable messages sending messages to at most $C$ factor nodes each, each with a cost of $O(np)$. Thus, the total cost of processing all variable nodes is $O(C^2 2^b + C^2 2^b np)$. The total cost of message is dominated by processing the factors.

The total complexity is then $O(2^b(n_c t + t^2 + n(p + b))$. Note that $p = n_c - n = t \log(n_c)$. Due to the structure of the code and the relationship between $n, p$ and $n_c$, one could stop here, and it would be best to if we want to consider very large $t$. For the purposes of exposition, we will assume that $t \ll n$, which implies $n > p$, and thus $p \approx t \log(n)$. In this case, we can write:

$$\text{Complexity} = O(2^b(nt \log(n) + nb)) \tag{25}$$

To arrive at the stated equation in Section 1, we take $2^b = O(s)$. Under the low degree assumption, we have $s = O(d \log(n))$. Then assuming we take $t = O(d)$, we arrive at a complexity of $O(sdn \log(n))$.

## B. Experiment Details

### B.1. Implementation Details

Experiments are run on a server using Nvidia L40S GPUs and A100 GPUs. When splitting text into words or sentences, we make use of the default word and sentence tokenizer from `nltk` (Bird et al., 2009). To fit regressions, we use the `scikit-learn` (Pedregosa et al., 2011) implementations of `LinearRegression` and `RidgeCV`.

### B.2. Datasets and Models

B.2.1. SENTIMENT ANALYSIS

152 movie reviews were used from the *Large Movie Review Dataset* (Maas et al., 2011), supplemented with 8 movie reviews from the *Stanford Sentiment Treebank* dataset (Socher et al., 2013). These 160 reviews were categorized using their word counts into 8 groups ([8-15, 16-32, ..., 1024-2047]), with 20 reviews in each group.

To measure the sentiment of each movie review, we utilize a `DistilBERT` model (Sanh et al., 2019) fine-tuned for sentiment analysis (Odabasi, 2025). When masking, we replace the word with the `[UNK]` token. We construct an value function over the output logit associated with the positive class.

B.2.2. HOTPOTQA

We consider 100 examples from the *HotpotQA*(Yang et al., 2018) dataset. These examples were categorized using the number of sentences into four groups ([8-15, 16-32, 32-64, 64-127]). We use a `Llama-3.2-3B-Instruct` model with

8-bit quantization. When masking, we replace with the `[UNK]` token, and measure the log-perplexity of generating the original output. Since *HotpotQA* is a multi-document dataset, we use the following prompt format.

> **Title:** {title_1}
> **Content:** {document_1}
> . . .
> **Title:** {title_m}
> **Content:** {document_m}
>
> **Query:** {question}. Keep your answers as short as possible.

### B.2.3. DROP

We consider 100 examples from the *DROP* (Yang et al., 2018) dataset. These examples were categorized using the number of words into six groups ([8-15, 16-32, 32-64, 64-127, 128-256, 512-1024]). We use the same model as *HotpotQA* and mask in a similar fashion. We use the following prompt format.

> **Context:** {context}
> **Query:** {question}. Keep your answers as short as possible.

### B.2.4. TROLLEY PROBLEM

The simplified trolley problem was inspired by the one provided in (Williams & Huckle, 2024). When masking, the `[UNK]` token was used to replace words. The following prompt was given to `gpt-4o-mini-2024-07-18`:

> **System:** Answer with the one word True or False only. Any other answer will be marked incorrect.
> **User:** {Masked Input} True or False: You should not pull the lever.

A value function was created by finding the difference between the model's logprob associated with the "True" token minus the logprob of the "False" token.

### B.2.5. VISUAL QUESTION ANSWERING

The base image was partitioned into a $6 \times 8$ grid. To mask, Gaussian blur was applied to the masked cells. The masked image was input into `LLaVA-NeXT-Mistral-7B`, a large multimodal model, with the following prompt:

> **Context:** {masked image}
> **Query:** What is shown in this image?

The original output to the unmasked image is "A dog playing with a basketball." Using the masked images, we build a value function that measures the probability of generating the original output sequence (log probability).

### B.3. Baselines

The following marginal feature attribution baselines were run:

1. *LIME*: LIME (Local Interpretable Model-agnostic Explanations) (Ribeiro et al., 2016) uses LASSO to build a sparse linear approximation of the value function. The approximation is weighted to be *local*, using an exponential kernel to fit the function better closer to the original input (less maskings).

2. *SHAP*: Implemented using KernelSHAP (Lundberg & Lee, 2017), SHAP interprets the value function as a cooperative game and attributes credit to each of the features according to the Shapley value. KernelSHAP approximates the Shapley

values of this game through solving a weighted least squares problem, where the weighting function is informed by the Shapley kernel, promoting samples where very either very few or most inputs are masked.

3. *Banzhaf*: Similar to Shapley values, Banzhaf values (Banzhaf III, 1964) represent another credit attribution concept from cooperative game theory. We compute the Banzhaf values by fitting a ridge regression to uniformly drawn samples, selecting the regularization parameter through cross-validation.

Furthermore, we compared against the following interaction attribution methods:

4. *Faith-Banzhaf*: The Faith-Banzhaf Interaction Index (Tsai et al., 2023), up to degree $t$, provides the most faithful $t^{\text{th}}$ order polynomial approximation of the value function under a uniform kernel. We obtain this approximation using cross-validated ridge regression on uniformly drawn samples.

5. *Faith-Shap*: Similarly, the Faith-Shapley Interaction Index (Tsai et al., 2023), up to degree $t$, provides the most faithful $t$ order polynomial approximation of the value function under a Shapley kernel. As described in (Tsai et al., 2023), the indices can be estimated through solving a weighted least squares problem. We use the implementation provided in SHAP-IQ (Muschalik et al., 2024).

6. *Shapley-Taylor*: The Shapley-Taylor Interaction Index (Sundararajan et al., 2020), up to degree $t$, provides another interaction definition based on the Taylor Series of the Möbius transform of the value function. To estimate the interaction indices, we leverage the sample-efficient estimator SVARM-IQ (Kolpaczki et al., 2024), as implemented in SHAP-IQ (Muschalik et al., 2024).

### B.4. Sample Complexity

The total number of samples needed for SPEX is $\approx C2^b t \log(n)$. We fix $C = 3$ and $t = 5$. The table below presents the number of samples used in our experiments for various choices of sparsity parameter $b$ and different input sizes $n$:

| Sparsity Parameter ($b$) | Number of Inputs ($n$) | | | | | | |
|---|---|---|---|---|---|---|---|
| | 8–11 | 12–36 | 37–92 | 93–215 | 216–466 | 467–973 | 974–1992 |
| 4 | 1,008 | 1,344 | 1,728 | 1,968 | 2,208 | 2,448 | 2,688 |
| 6 | 4,032 | 5,376 | 6,912 | 7,872 | 8,832 | 9,792 | 10,752 |
| 8 | 16,128 | 21,504 | 27,648 | 31,488 | 35,328 | 39,168 | 43,008 |

Table 2: Number of samples needed for each $b$ and $n$.

### B.5. Additional Results

#### B.5.1. FAITHFULNESS

**Faithfulness for a fixed sparsity parameter** $b = 8$: We first measure the faithfulness by scaling the number of samples logarithmically with $n$. The exact number of samples used can be found in Table 2.

In Table 3, we showcase the average faithfulness of every runnable method across every group of examples for the *Sentiment*, *DROP*, and *HotpotQA* datasets. Among marginal attribution methods, LIME and Banzhaf achieve the best faithfulness. SHAP's faithfulness worsens as $n$ grows, though this is unsurprising, as Shapley values are only efficient (intended to sum to the unmasked output), not faithful.

Comparing to interaction methods, SPEX is comparable to the highest order Faith-Banzhaf that can feasible be run at every size of $n$. However, due to poor computation complexity scaling of this, and other interaction methods, these methods are only able to be used for small $n$. In particular, we found Shapley-Taylor difficult to run for the *DROP* and *HotpotQA* tasks, unable to finish within thirty minutes.

We also report the average sparsity and the average sparsity ratio (sparsity over total interactions) discovered by SPEX for each of the groups. For *Sentiment*, once reaching the $n \in [128 - 255]$ group, the average sparsity is found to be less than the number of inputs! Yet, SPEX is still able to achieve high faithfulness and significantly outperform linear methods like LIME and Banzhaf.

| | n | Avg. Sparsity | Avg. Sparsity Ratio | SPEX | LIME | Banzhaf | Faith-Banzhaf | | | SHAP | Faith-Shap | | | Shapley-Taylor | | |
|---|---|---|---|---|---|---|---|---|---|---|---|---|---|---|---|---|
| | | | | | | | 2nd | 3rd | 4th | | 2nd | 3rd | 4th | 2nd | 3rd | 4th |
| *Sentiment* | 8-15 | 369.9 | $3.70 \times 10^{-1}$ | 1.00 | 0.83 | 0.84 | 0.96 | 0.99 | 1.00 | 0.62 | 0.93 | 0.98 | 1.00 | 0.72 | 0.81 | 0.92 |
| | 16-31 | 208.7 | $2.31 \times 10^{-4}$ | 0.97 | 0.75 | 0.75 | 0.93 | 0.98 | | 0.20 | 0.89 | 0.95 | | 0.46 | -710.85 | |
| | 32-63 | 149.7 | $3.14 \times 10^{-9}$ | 0.93 | 0.67 | 0.68 | 0.90 | | | -0.25 | 0.82 | | | -0.30 | | |
| | 64-127 | 118.4 | $2.19 \times 10^{-19}$ | 0.87 | 0.66 | 0.66 | | | | -1.17 | | | | | | |
| | 128-255 | 113.5 | $1.40 \times 10^{-38}$ | 0.82 | 0.63 | 0.63 | | | | -3.94 | | | | | | |
| | 256-511 | 100.4 | $5.48 \times 10^{-78}$ | 0.76 | 0.62 | 0.62 | | | | -6.77 | | | | | | |
| | 512-1013 | 86.1 | $2.02 \times 10^{-155}$ | 0.73 | 0.61 | 0.61 | | | | -4.78 | | | | | | |
| | 1024-2047 | 81.7 | $3.78 \times 10^{-302}$ | 0.71 | 0.59 | 0.59 | | | | -17.31 | | | | | | |
| *DROP* | 32-63 | 77.1 | $1.97 \times 10^{-14}$ | 0.58 | 0.29 | 0.29 | 0.53 | | | -0.07 | 0.17 | | | N/A | | |
| | 64-127 | 56.3 | $2.59 \times 10^{-24}$ | 0.51 | 0.30 | 0.30 | | | | 0.02 | | | | | | |
| | 128-255 | 58.7 | $2.31 \times 10^{-38}$ | 0.67 | 0.43 | 0.43 | | | | 0.14 | | | | | | |
| | 256-511 | 56.7 | $1.29 \times 10^{-76}$ | 0.48 | 0.43 | 0.44 | | | | -5.29 | | | | | | |
| | 512-1023 | 36.3 | $9.63 \times 10^{-155}$ | 0.55 | 0.46 | 0.48 | | | | -0.23 | | | | | | |
| *HotpotQA* | 8-15 | 108.3 | $1.10 \times 10^{-2}$ | 0.87 | 0.38 | 0.38 | 0.63 | 0.77 | 0.84 | -1.09 | -19.14 | -2.33 | 0.73 | N/A | N/A | N/A |
| | 16-31 | 96.4 | $3.30 \times 10^{-4}$ | 0.67 | 0.39 | 0.49 | 0.66 | 0.72 | | 0.23 | -2.28 | 0.40 | | N/A | N/A | |
| | 32-63 | 79.4 | $5.86 \times 10^{-9}$ | 0.57 | 0.44 | 0.45 | 0.59 | | | 0.13 | 0.32 | | | N/A | | |
| | 64-127 | 73.0 | $1.02 \times 10^{-18}$ | 0.63 | 0.53 | 0.53 | | | | -0.31 | | | | | | |

Table 3: Faithfulness across all baseline methods for fixed $b = 8$. The average recovered sparsity and the average ratio between sparsity and total possible interactions is also reported.

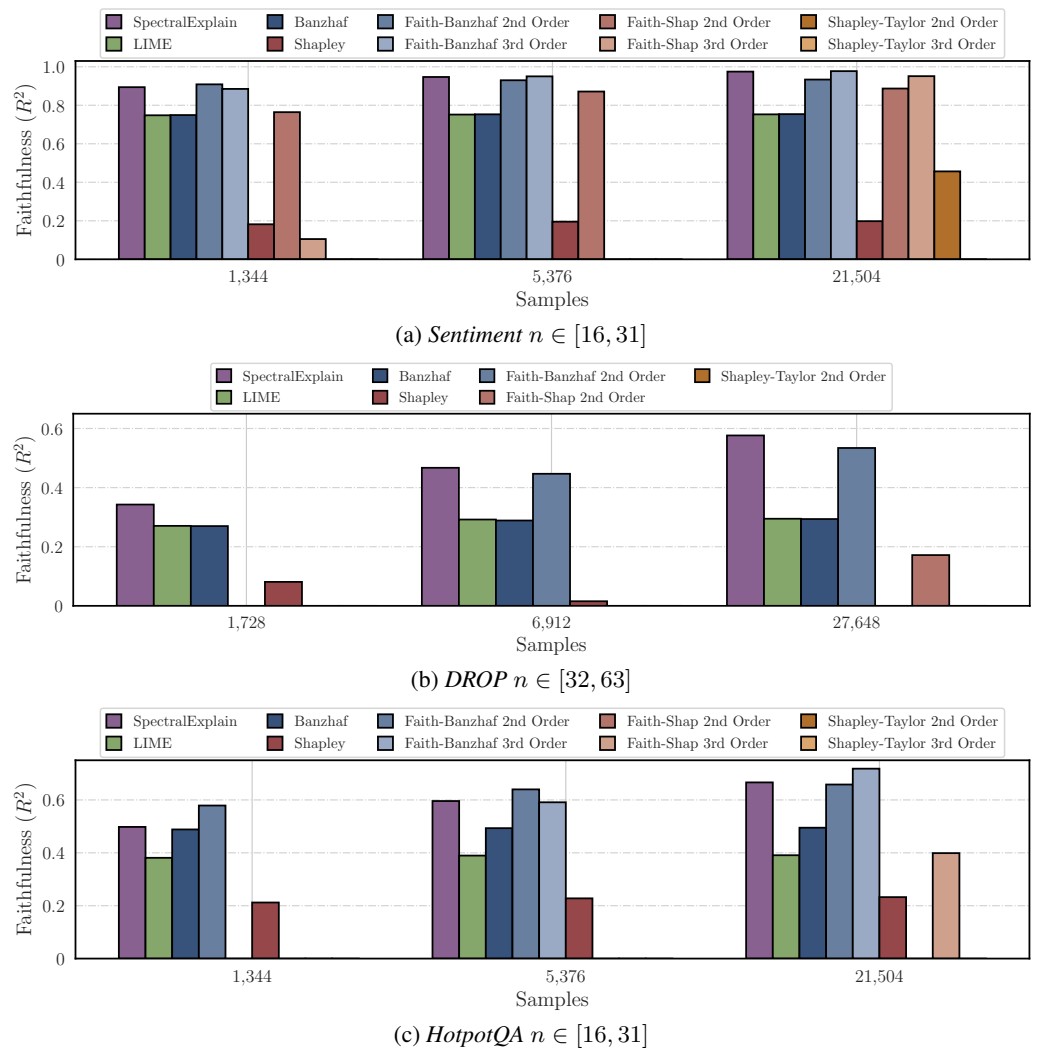

(a) *Sentiment* $n \in [16, 31]$

(b) *DROP* $n \in [32, 63]$

(c) *HotpotQA* $n \in [16, 31]$

Figure 9: Faithfulness across the three datasets for all methods that can be feasibly ran. Methods appearing in the legend, but not in the plot had a faithfulness below 0 for the given number of samples.

**Faithfulness for varying the sparsity parameter** $b$**:** $b = 8$ may not be the sparsity parameter that achieves the best trade-off between samples and faithfulness. For instance, with complex generative models, the cost or time per instance may necessitate taking fewer samples.

In Fig. 9, we showcase the faithfulness results for SPEX and all baseline methods when $b$ is $4, 6, 8$. Since the samples taken by the algorithm grows with $2^b$, $b = 8$ takes 16 times more samples than $b = 4$. Even in the low-sample regime, SPEX achieves high faithfulness, often surpassing linear models and second order models. At this scale, we find that third and fourth order models often do not have enough samples to provide a good fit.

**Hyperparameter selection of** $b, t,$ **and** $C$**:** We conduct hyperparameter ablation for the *sentiment analysis* task, evaluating faithfulness on a held-out set of 10,000 masks, averaged across 20 movie reviews per group. The number of training samples collected follows the approximation $\approx C 2^b t \log(n)$. By default, we set $b = 8$, $C = 3$, and $t = 5$, varying each parameter independently. The first plot shows that $b = 8$ strikes a balance between training efficiency and faithfulness, while the second indicates that performance stabilizes beyond $C = 3$. Finally, we observe that although some movie reviews exhibit strong higher-order interactions, maximum faithfulness is achieved with $t = 4$ or $t = 5$.

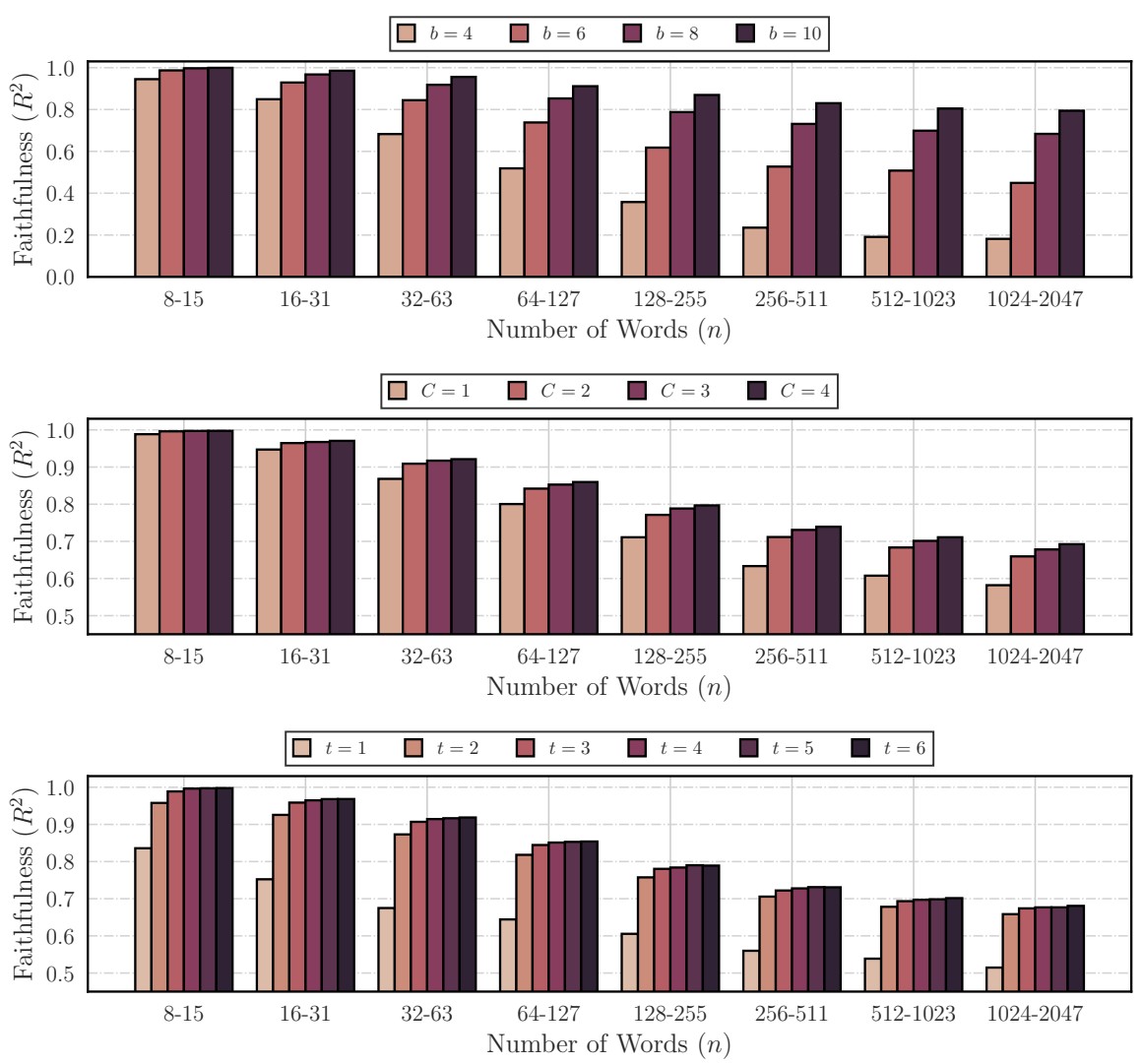

Figure 10: Hyperparameter comparison for the *sentiment analysis* task.

B.5.2. ABSTRACT REASONING

We also evaluated the performance of `Llama 3.2 3B-Instruct` (Grattafiori et al., 2024) on the modified trolley problem. As a reminder, the modified problem is presented below:

> A runaway trolley is heading **away** from five people who are tied to the track and cannot move. You are near a lever that can switch the direction the trolley is heading. Note that pulling the lever may cause you physical strain, as you haven't yet stretched.
>
> **True** or **False**: **You should not pull the lever.**

Across 1,000 evaluations, the model achieves an accuracy of just 11.8%. Despite a similar accuracy to `GPT-4o mini`, the SHAP and SPEX-computed interactions indicate that the two models are lead astray by different parts of the problem.

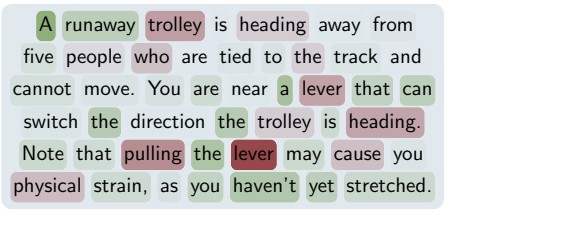

(a) SHAP values        (b) Interactions computed via SPEX

Figure 11: SHAP and SPEX-computed interactions computed for `Llama 3.2 3B-Instruct`'s answering of the modified trolley problem. Words and interactions highlighted in green contribute positively to producing the correct output, while those in red lead the model toward an incorrect response.

The most negative SHAP values of `Llama 3.2 3B-Instruct` appear for later terms such as *pulling* and *lever*, with surrounding words having positive SHAP values. The SPEX-computed interactions tell a different story; many of the words in the last sentence have a negative first order value, with a significant third order interaction between *you haven't yet*. Furthermore, the first word *A* possesses a strong negative second order interaction with *trolley*. Although counterintuitive— since the fact about stretching should only enhance the likelihood of a correct answer—removing the non-critical final sentence unexpectedly boosts the model's accuracy to 20.8%, a 9% improvement.

## C. Relationships between Fourier and Interaction Concepts

**Fourier to Möbius Coefficients:** The Möbius Coefficients, also referred to as the *Harsanyi dividends*, can be recovered through (Grabisch, 2016):

$$I^M(S) = (-2)^{|S|} \sum_{T \supseteq S} F(T). \tag{26}$$

**Fourier to Banzhaf Interaction Indices:** Banzhaf Interactions Indices (Roubens, 1996) have a close relationship to Fourier coefficients. As shown in (Grabisch et al., 2000):

$$I^{BII}(S) = \sum_{T \supseteq S} \frac{2^{|S|}}{2^{|T|}} I^M(T). \tag{27}$$

Using the relationship from Eq. 26,

$$I^{BII}(S) = \sum_{T \supseteq S} \frac{2^{|S|}}{2^{|T|}} (-2)^{|T|} \sum_{R \supseteq T} F(R) \tag{28}$$

$$= 2^{|S|} \sum_{T \supseteq S} (-1)^{|T|} \sum_{R \supseteq T} F(R) \tag{29}$$

$$= 2^{|S|} \sum_{R \supseteq S} F(R) \sum_{S \subseteq T \subseteq R} (-1)^{|T|}, \tag{30}$$

$$= (-2)^{|S|} F(S) \tag{31}$$

where the last line follows due to $\sum_{S \subseteq T \subseteq R} (-1)^{|T|}$ evaluating to 0 unless $R = S$.

When $S$ is a singleton, we recover the relationship between Fourier Coefficients and the Banzhaf Value $BV(i)$:

$$BV(i) = I^{BII}(\{i\}) = -2F(\{i\}). \tag{32}$$

**Fourier to Shapley Interaction Indices:** Shapley Interaction Indices (Grabisch, 1997) are a generalization of Shapley values to interactions. Using the following relationship to Möbius Coefficients (Grabisch et al., 2000):

$$I^{SII}(S) = \sum_{T \supseteq S} \frac{I^M(S)}{|T| - |S| + 1} \tag{33}$$

$$= \sum_{T \supseteq S} \sum_{R \supseteq T} \frac{(-2)^{|T|} F(R)}{|T| - |S| + 1} \tag{34}$$

$$= \sum_{R \supseteq S} F(R) \sum_{S \subseteq T \subseteq R} \frac{(-2)^{|T|}}{|T| - |S| + 1} \tag{35}$$

$$= \sum_{R \supseteq S} F(R) \sum_{j=|S|}^{|R|} \frac{(-2)^j}{j - |S| + 1} \binom{|R| - |S|}{j - |S|} \tag{36}$$

$$= \sum_{R \supseteq S} F(R) \sum_{k=0}^{|R|-|S|} \frac{(-2)^{k+|S|}}{k + 1} \binom{|R| - |S|}{k} \tag{37}$$

$$= (-2)^{|S|} \sum_{R \supseteq S} F(R) \sum_{k=0}^{|R|-|S|} \frac{(-2)^k}{k + 1} \binom{|R| - |S|}{k} \tag{38}$$

Consider the following integral and an application of the binomial theorem:

$$\int_0^t (1 + x)^{|R|-|S|} dx = \int_0^t \sum_{k=0}^{|R|-|S|} \binom{|R| - |S|}{k} x^k dx \tag{39}$$

$$= \sum_{k=0}^{|R|-|S|} \binom{|R| - |S|}{k} \int_0^t x^k dx \tag{40}$$

$$= \sum_{k=0}^{|R|-|S|} \binom{|R| - |S|}{k} \left( \frac{t^{k+1}}{k + 1} \right) \tag{41}$$

Evaluating at $t = -2$:

$$\sum_{k=0}^{|R|-|S|} \binom{|R|-|S|}{k} \left( \frac{(-2)^k}{k+1} \right) = -\frac{1}{2} \int_0^{-2} (1+x)^{|R|-|S|} dx \tag{42}$$

$$= -\frac{1}{2} \cdot \frac{(-1)^{|R|-|S|+1} - 1}{|R| - |S| + 1} \tag{43}$$

$$= \begin{cases} \frac{1}{|R|-|S|+1}, & \text{if Parity}(|R|) = \text{Parity}(|S|) \\ 0, & \text{otherwise} \end{cases} \tag{44}$$

As a result, we find the relationship between Shapley Interaction Indices and Fourier Coefficients:

$$I^{SII}(S) = (-2)^{|S|} \sum_{\substack{R \supseteq S, \\ (-1)^{|R|} = (-1)^{|S|}}} \frac{F(R)}{|R| - |S| + 1}. \tag{45}$$

When $S$ is a singleton, we recover the relationship between Fourier Coefficients and the Shapley Value $SV(i)$:

$$SV(i) = I^{SII}(\{i\}) = (-2) \sum_{\substack{R \supseteq \{i\}, \\ |R| \text{ is odd}}} \frac{F(R)}{|R|}. \tag{46}$$

**Fourier to Faith-Banzhaf Interaction Indices:** Faith-Banzhaf Interaction Indices (Tsai et al., 2023) of up to degree $\ell$ are the unique minimizer to the following regression objective:

$$\sum_{S \subseteq [n]} \left( f(S) - \sum_{T \subseteq S, |T| \leq \ell} I^{FBII}(T, \ell) \right)^2. \tag{47}$$

Let $g(S)$ be the XOR polynomial up to degree $\ell$ that minimizes the regression objective. Appealing to Parseval's identity,

$$\sum_{S \subseteq [n]} (f(S) - g(S))^2 = \sum_{S \subseteq [n]} (F(S) - G(S))^2 = \sum_{S \subseteq [n], |S| \leq \ell} (F(S) - G(S))^2 + \sum_{S \subseteq [n], |S| > \ell} F(S)^2, \tag{48}$$

which is minimized when $G(S) = F(S)$ for $|S| \leq \ell$. Using Eq. 26, it can be seen that the Faith-Banzhaf Interaction Indices correspond to the Möbius Coefficients of the function $f(S)$ truncated up to degree $\ell$:

$$I^{FBII}(S, \ell) = (-2)^{|S|} \sum_{T \supseteq S, |T| \leq \ell} F(T). \tag{49}$$

**Fourier to Faith-Shapley Interaction Indices:** Faith-Shapley Interaction Indices (Tsai et al., 2023) of up to degree $\ell$ have the following relationship to Möbius Coefficients:

$$I^{FSII}(S, \ell) = I^M(S) + (-1)^{\ell-|S|} \frac{|S|}{\ell+|S|} \binom{\ell}{|S|} \sum_{T \supset S, |T| > \ell} \frac{\binom{|T|-1}{\ell}}{\binom{|T|+\ell-1}{\ell+|S|}} I^M(T) \tag{50}$$

$$= (-2)^{|S|} \sum_{T \supseteq S} F(T) + (-1)^{\ell-|S|} \frac{|S|}{\ell+|S|} \binom{\ell}{|S|} \sum_{T \supset S, |T| > \ell} \frac{\binom{|T|-1}{\ell}}{\binom{|T|+\ell-1}{\ell+|S|}} (-2)^{|T|} \sum_{R \supseteq T} F(R) \tag{51}$$

$$= (-2)^{|S|} \sum_{T \supseteq S} F(T) + (-1)^{\ell-|S|} \frac{|S|}{\ell+|S|} \binom{\ell}{|S|} \sum_{R \supset S, |R| > \ell} F(R) \sum_{S \subset T \subseteq R, |T| > \ell} \frac{\binom{|T|-1}{\ell}}{\binom{|T|+\ell-1}{\ell+|S|}} (-2)^{|T|}. \tag{52}$$

**Fourier to Shapley-Taylor Interaction Indices:** Shapley-Taylor Interactions Indices (Sundararajan et al., 2020) of up to degree $\ell$ are related to Möbius Coefficients in the following way:

$$I^{STII}(S, \ell) = \begin{cases} I^M(S), & \text{if } |S| < \ell \\ \sum_{T \supseteq S} \binom{|T|}{\ell}^{-1} I^M(T), & \text{if } |S| = \ell. \end{cases} \tag{53}$$

From an application of Eq. 26,

$$I^{STII}(S, \ell) = \begin{cases} (-2)^{|S|} \sum_{T \supseteq S} F(T), & \text{if } |S| < \ell \\ \sum_{T \supseteq S} \binom{|T|}{\ell}^{-1} (-2)^{|T|} \sum_{R \supseteq T} F(R), & \text{if } |S| = \ell. \end{cases} \tag{54}$$

Simplifying the sum in the $|S| = \ell$ case:

$$\sum_{T \supseteq S} \binom{|T|}{\ell}^{-1} (-2)^{|T|} \sum_{R \supseteq T} F(R) = \sum_{R \supseteq S} F(R) \sum_{S \subseteq T \subseteq R} \binom{|T|}{\ell}^{-1} (-2)^{|T|} \tag{55}$$

$$= \sum_{R \supseteq S} F(R) \sum_{k=\ell}^{|R|} \binom{k}{\ell}^{-1} (-2)^k \binom{|R| - \ell}{k - \ell} \tag{56}$$

$$\tag{57}$$

Hence,

$$I^{STII}(S, \ell) = \begin{cases} (-2)^{|S|} \sum_{T \supseteq S} F(T), & \text{if } |S| < \ell \\ \sum_{T \supseteq S} F(T) \sum_{k=\ell}^{|T|} \binom{k}{\ell}^{-1} (-2)^k \binom{|T| - \ell}{k - \ell}, & \text{if } |S| = \ell. \end{cases} \tag{58}$$

