# OpenReview forum: "SPEX: Scaling Feature Interaction Explanations for LLMs"
_ICML.cc/2025/Conference — ICML 2025 poster_

### Official Review · Reviewer_ehfY · 2025-02-26

**Overall Recommendation:** 4

**Summary:**

This paper introduces SPEX, a model-agnostic interaction attribution algorithm designed to scale feature interaction explanations to large input spaces, e.g., LLMs. The key contribution of SPEX is leveraging a sparse Fourier transform with channel decoding to efficiently identify and reconstruct important feature interactions. Since the existing methods for interaction explanations do not scale beyond small input sizes (e.g., Faith-Shap, SHAP-IQ), SPEX is the only method that can scale to inputs of up to ~1000 while improving faithfulness in reconstructing LLM outputs. Experiments are conducted on three long-context datasets: Sentiment Analysis, HotpotQA, and DROP, showing that SPEX outperforms baselines.

## update after rebuttal

The author's response to my question regarding mechanistic interpretability is still relatively vague and high-level, but I think it is reasonable given this paper is a feature explanation paper. I would encourage the authors to discuss different paradigms of explanation in the paper, as it is not quite clear what the differences and advantages a newly proposed method has compared to methods from a different perspective. I have raised my score to 4.

**Claims And Evidence:**

Yes.

- The claim of SPEX being efficient and scalable to large input sizes is supported both theoretically by the time complexity of O(sdn) and empirical runtime comparisons of different imput sizes (Figure 5).

- There are also experiments supporting SPEX's faithfulness on Sentiment, QA datasets and human-labeled annotations.

**Essential References Not Discussed:**

The coverage of related work is reasonable. The XAI literature is too huge to cover comprehensively.

More discussions about the relationship to the recent Mechanistic Interpretability domain would be better, given this domain mainly focuses on LLM explanation.

**Experimental Designs Or Analyses:**

Yes, the experiments design is reasonable with comparison against a broad set of baselines, including marginal attributions (LIME, SHAP, Banzhaf) and interaction indices (Faith-Shap, Faith-Banzhaf, Shapley-Taylor) on three datasets for different tasks.

**Methods And Evaluation Criteria:**

Yes. The method is evaluated in terms of Faithfulness, Feature Removal, Recovery@10 and some case studies.

**Other Comments Or Suggestions:**

No.

**Other Strengths And Weaknesses:**

Strengths

- SPEX addresses the key scalability limitation of prior interaction attribution methods.

- The case studies are another strength, which demonstrates real-world relevance by debugging LLM reasoning failures and VQA tasks.

Weaknesses

- Limited model diversity. The experiments focus primarily on LLaMA-3 and GPT-4o Mini. Further evaluations on other LLMs would strengthen generalizability.

- While SPEX is efficient, its robustness under noisy or adversarial perturbations remains unclear. A discussion regarding this perspective would be better.

**Questions For Authors:**

In the era of LLMs, how useful are the feature attribution methods, especially compared to directly asking LLM to provide a reason or explanation of its output?

**Relation To Broader Scientific Literature:**

The paper is related to the  XAI literature. It builds upon: marginal feature attribution methods and feature interaction attributions

It also makes connections to sparse Fourier transforms, which is relatively underexplored in XAI, making this a novel contribution.

**Theoretical Claims:**

Yes, theoretical claims regarding Fourier transform are discussed in App A.  I scanned through the App A. Seems to be correct.

---

> ### Author Rebuttal · Authors · 2025-04-01
>
> **This rebuttal contains (anonymized) links to figures. We also built a web app to help explore SPEX: https://anon858023.github.io/spex-webapp/**
>
> Thank you for the review. We hope that with the proposed additions based on your comments, your concerns are addressed and we can convince you that this manuscript is not a borderline case.
>
> **Discussion of Mechanistic Interpretability** We will add a discussion of mechanistic interpretability, which is a very timely topic in the space of LLMs. Specifically, much work in mechanistic interpretability focuses on discovering sparse circuits. Alternatively, this can be seen as finding interactions between attention heads (i.e., features in the context of our paper). Interesting future work can explore the application of SPEX to sparse feature discovery and other topics in mechanistic interpretability.
>
> **Importance of feature attribution with LLMs** Our approach offers several key advantages over asking an LLM to explain itself.
>
> 1. Many transformer-based models are encoder only which are unable to generate such explanations. Our experiments on the sentiment-analysis demonstrate the strong performance on SPEX on these models. Further, generative models such as protein language models are also unable to generate such explanations.
> 2. Recent work has shown that LLMs often cannot explain themselves (https://arxiv.org/pdf/2405.04382), and self-explanations do not accurately reflect the underlying reasoning process of the model. On the other hand, our approach is grounded in the model output, and can be systematically verified via our faithfulness and top-$r$ removal experiments, while LLM explanations are not.
>
> **More models** We have considered 4 different models in this work: DistillBERT, Llama-3.2, GPT 4o-mini and LLaVA-NeXT-Mistral-7B. For additional diversity, we repeat the faithfulness experiments with two additional models: Qwen2-7B-Instruct and Mistral-7B on the DROP dataset, and provide the results in the table below. Due to limited time, we only re-ran SPEX and the Faith-Banzhaf methods, and consider $5$ examples for each regime of $n$. We chose Faith-Banzhaf methods since they displayed the most competitive performance with SPEX in our previous experiments.
>
> | Model                | n       | SPEX  | Banzhaf | Faith-Banzhaf 2nd |
> |----------------------|---------|------:|--------:|----------------:|
> | *Qwen2-7B-Instruct* | 32–63   | **0.850** | 0.483  | 0.727          |
> |                      | 64–127  | **0.559** | 0.441  | —              |
> |                      | 128–255 | **0.670** | 0.513  | —              |
> |                      | 256–511 | **0.549** | 0.445  | —              |
> | *Mistral-7B*        | 32–63   | **0.931** | 0.864  | 0.904          |
> |                      | 64–127  | **0.822** | 0.691  | —              |
> |                      | 128–255 | **0.834** | 0.574  | —              |
> |                      | 256–511 | **0.853** | 0.542  | —              |
>
> *Table*: Faithfulness on DROP dataset across input lengths (*n*) for Mistral-7B and Qwen2-7B-Instruct.
>
> We continue to achieve significantly higher faithfulness across models. We will include these results, and run them for the other interpretability methods as well as for HotpotQA in the camera-ready version.
>
> **Noise Robustness** We find that the use of the BCH code makes SPEX impressively robust. We've conducted an experiment where we repeat our Sentiment experiment with additional observation noise. Our results demonstrate the robustness of SPEX to noise, as it mostly retains its $R^2$ performance even under very high noise levels. (https://imgur.com/a/qAf0y2J). These robustness results will be included in the camera-ready version of the paper.

---

> > ### Comment · Reviewer_ehfY · 2025-04-02
> >
> > Thank the authors for their response. To follow up, I would like to find out the authors' view on mechanistic interpretability vs. feature attribution. Can the authors extend the discussion about mechanistic interpretability a little bit?
> >
> > For example, are they fundamentally the same and unified? Or are they more distinct? Can they complement each other and be combined together to achieve more coherent and convincing explanations of language models? If so, any more concrete ideas?

---

> > > ### Author Response · Authors · 2025-04-04
> > >
> > > Mechanstic Interpretability is an exciting and active research direction, with many different relevant sub-areas. There has also been a recent effort to unify approaches between mechanistic interpretability and feature attribution. [i] provides a good summary of these efforts. In particular SPEX belongs to the class of *model agnostic* approaches.  These model agnostic approaches don't require the *inputs* to correspond to the natural inputs of the model.
> > >
> > > When we have access to model internals, we could apply SPEX to identify important model components, where the *inputs* to the value function correspond to different model components and *masking patterns* represent the ablation of subsets of model components. Similar approaches (using marginal attribution methods) were used in [ii] and [iii].
> > >
> > > **Here are some of our concrete thoughts on how we might combine and connect some ideas (below) from SPEX to Mechantistic Interpretability:**
> > > 1. *Structural observation of a sparse, low-degree structure in feature space.*
> > > 2. *The mathematical construction SPEX uses to exploit this structure.*
> > >
> > > **Component/Neuron Attribution/ Localization:** This is perhaps the most straightforward connection to (**2**). If sufficient sparsity exists in the space of neurons or components, we can potentially use algebraic tools to carefully perturb these components and find a sparse set of them most relevant to a specific task.
> > >
> > > **Sparse Semantic Linear Representation of Activations, (SAEs, Transcoders, Crosscoders, etc.):** This branch of the literature is also primarily concerned with sparsity, not in feature interaction space, i.e., (**1**), but in **activation space**. We think the sparsity in feature interaction space and activation space can be *complimentary*. For example, features with strong interactions according to SPEX may result in activations with correlated encodings. One might even think of observing a matrix $\mathbf{X} \in \mathbb{R}^{d \times n}$ where $d$ is activation dimension and $n$ is the context length, and jointly applying dictionary learning ideas (like SAEs) and Fourier transforms to exploit the additional structure. There are a whole host of interesting things you can do by combining these notions of sparsity:
> > >
> > > - More sparsity $\implies$ more structure $\implies$ potential for greater efficiency in training things like encoders.
> > > - Joint analysis of sparsity. If we can jointly grasp semantic representations and interactions in feature space, we might be able to understand not just the presence of interactions but also, which semantics are involved in these interactions.
> > >
> > > *We think this is just scratching the surface of potential applications, and hope that the publication of this work can also have impact on the literature of Mechanistic Interpretability.*
> > >
> > > ***We thank you again for your time. We hope you will consider this discussion and our thorough answers to your other questions in your final review score.***
> > >
> > >
> > >
> > > **References:**
> > >
> > > [i] Zhang, Shichang, et al. "Building Bridges, Not Walls--Advancing Interpretability by Unifying Feature, Data, and Model Component Attribution."
> > >
> > > [ii] Ghorbani, Amirata, et al. "Neuron shapley: Discovering the responsible neurons."
> > >
> > > [iii] Shah, Harshay, et al. "Decomposing and editing predictions by modeling model computation."

---

### Official Review · Reviewer_e6K4 · 2025-03-11

**Overall Recommendation:** 4

**Summary:**

This paper proposes a new algorithm to efficiently compute the sparse Fourier transform and identify salient interactions by leveraging the underlying sparse structure of interactions. The proposed methods outperform previous attribution methods and interaction indices in faithfulness measure while costing much less computation time.

**Claims And Evidence:**

The claims are supported by corresponding evidence.

**Essential References Not Discussed:**

No essential references not discussed.

**Experimental Designs Or Analyses:**

1.	There is no validation or ablation study on how the hyperparameters are set (e.g., $C=3$, $t=5$, $b=8$). I suggest the authors clarify the reason for choosing the specific set of hyperparameters or conduct extra experiments to show the effects of different hyperparameters.

2.	In Figure 5, with the increase of the number of input variables $n$, the SPEX method exhibits decreasing faithfulness but with almost identical compute time. Is it possible to obtain higher faithfulness with a slight degradation in compute efficiency (e.g., by tuning some of the hyperparameters in SPEX)?

3.	The faithfulness is measured on 10000 randomly sampled masks. However, the number 10000 is still small compared to the total number of possible masks $2^n$, which can be huge when $n>20$. According to the sparsity structure of interactions, it is possible that none of the sampled masks cover significant interactions. As a result, the output of value functions on these masks will be close to zero, and the evaluation metric will not make sense. I suggest the authors provide guarantee that such scenario will not occur, or clarify that such scenario is rare in real experiments.

**Methods And Evaluation Criteria:**

It is not clear how the method in this paper is different from that of Kang et al., 2024, which is a main reference in this paper. Some techniques seem very similar. I suggest the authors clarify what is inherited from Kang et al., 2024, and what are new contributions in the Related Work or Algorithm section.

1.	The notion of *aliasing* and *subsampling* also appears in Kang et al., 2024. This seems to be a core technique to leverage the sparse structure of interactions and boost sample efficiency.

2.	The philosophy of the **Designing shifts** part is similar to that of the **singleton detection** algorithm in Kang et al., 2024.

3.	The **message passing** algorithm and the use of the bipartite graph is similar to the **graph peeling** algorithm in Kang et al., 2024.

Moreover, although this paper shows a similar framework with Kang et al., 2024., this paper is able to deal with a much longer input (up to $n=1000$ input variables) than Kang et al., 2024 (only up to $n=20$ input variables). It is not clear which part contribute most to this improvement. Is it the shift design? Or the message passing algorithm?

Regarding the evaluation criteria, the *faithfulness* metric defined in this paper is slightly different from that in Kang et al., 2024. Specifically, this paper subtracts the mean of value function outputs across all possible masks in the denominator. I would suggest the authors explain the intuition behind this change.

Kang et al. Learning to understand: Identifying interactions via the mobius transform. arXiv preprint arXiv:2401.13371, 2024.

**Other Comments Or Suggestions:**

In Line 133, the paper writes “we replace with the [MASK] token.” However, in Appendix, it shows that the [UNK] token is used. Which one is true?

**Other Strengths And Weaknesses:**

Strengths:

1.	The paper focuses on an important problem in deploying interaction-based explanations in LLMs, i.e., their scalability to a massive number of input tokens.

**Questions For Authors:**

1.	Among the interactions computed by the SPEX method, how many of them are significant? How many of them are positive and how many are negative? What is the proportion of positive-negative cancellation? It would help if the authors can provide such statistics under different values of $n$.

2.	This paper focuses on interactions derived through the Fourier transform (I will term it *Fourier interaction* for simplicity), which is different from widely-used interaction metrics such as the Mobius transform and Faith-shap interaction index.
I wonder how we can interpret the intuitive meaning of a specific Fourier interaction w.r.t. the set $S$? According to Kang et al., 2024, the Mobius transform $I^M(S)$ measures the additional effect to the network output due to the formation of a coalition $S$，and this effect cannot be achieved by any subset of the coalition S. Does the Fourier interaction have similar intuitive meanings?

Also, I would like to the authors to respond to my comments on **Methods And Evaluation Criteria** and **Experimental Designs Or Analyses**.

---

Post rebuttal: The authors address my concerns. I have raised my score accordingly.

**Relation To Broader Scientific Literature:**

This paper is related to the broader literature of explainable AI, especially feature attribution and interaction-based explanation method.

**Theoretical Claims:**

The theoretical claim about the sample complexity of the proposed algorithm is not carefully checked.

---

> ### Author Rebuttal · Authors · 2025-04-01
>
> **This rebuttal contains (anonymized) links to figures. Web app to explore SPEX: https://anon858023.github.io/spex-webapp/**
>
> Thank you for the thorough and helpful review. We hope our proposed additions and clarifications convince you that this paper is not a borderline case, but rather an impactful and important contribution.
>
> **Kang et al.** This is the main theoretical inspiration behind this work, showing that sparse transforms and signal processing ideas can be used to compute feature interactions. As the reviewer notes, however, there is a large gap between the practical performance of SPEX and the Sparse Möbius Transform (SMT). We will add a paragraph specifically to highlight the significant effort that enabled this.
> 1. SPEX uses the Fourier transform, which is orthonormal. This improves robustness by avoiding noise amplification of SMT in the transform domain.
> 2. The sampling procedure uses different code structures (1) random linear codes and (2) BCH codes, both contributing to robustness.
> 3. The BCH code is used as a source-channel code, leveraging samples to both compressively sense each $\mathbf{k}$ and build robustness.
> 4. We add a soft decoder that exploits the real-valued nature of logits, unlike SMT, which quantizes output ratios. Switching between soft and hard decoding allows trading compute for superior performance (see KaFj rebuttal).
> 5. The SPEX message passing procedure enables superior detection of interaction effects, as we can decode multitons (cases where non-zero coefficients collide) if there is enough difference in interaction magnitude. SMT does not decode multitons.
>
> In exchange for superior practical performance, SPEX has a more involved implementation and lacks SMT’s clean theoretical results.
>
> **Faithfulness**  Kang et al. defines $f$ as zero mean, so our evaluation is identical.
>
> ### **Experimental Design**
>
> **Hyperparameters** All hyper-parameters $C$, $t$ and $b$  control the sample complexity (i.e., number of sampled masks). As discussed in Section 5.1, the number of training masks is $2^bCt\log_2(n)$. Faithfulness is monotonic in all 3 parameters. Since $b$ has the largest effect, we measured faithfulness for $b\in \\{4,6,8 \\}$ for all three datasets in Fig 9. We outperform marginal approaches and are competitive with enumerative interaction indices. We perform additional hyper-parameter sweeps over $C$, $b$, and $t$ for the _sentiment analysis_ task over all possible regimes of $n$. Results in https://imgur.com/G4k4Q91 show $C=3$ and $t=5$ achieve a favorable trade-off in faithfulness and sample-complexity. We will include these experiments in the camera-ready version.
>
> **"Missed Interactions"** This question hits at a deep point! Every Fourier interaction is present in _every test mask_ (see eq. 1), and influences the output proportional to its magnitude. The direction (+/-) of influence changes depending on the mask. This is _not true_ of the Möbius transform, for which interactions may or may not be involved in the output depending on the mask. This property of the Fourier transform (deeply connected to the orthonormal property) is the reason why this approach works so well. Our revised paper will reflect this discussion.
>
> Additional experimental evidence supporting this:
> 1. **Variance of test responses** High variance in model outputs is observed across all datasets. For instance, in https://imgur.com/a/pagr97R, we show test response distributions for varying-length examples from _Sentiment_.  Moreover, the test response distribution demonstrates clear bias, further indicating that variance in test responses is not an artifact of random noise.
> 2. **Difference in performance across methods** If this hypothesis were true, all feature attribution methods would have the same faithfulness since they would be evaluated on test samples with $0$ variance.
> 3. **Removal Results** Our top-$r$ removal results indicate we learn meaningful information.
>
> **[MASK] vs [UNK]** Thank you, we will update this. [UNK] is used.
>
> **SPEX Interaction Statistics** We've conducted a series of experiments to evaluate this. See figure caption for details:
> 1. Significant interactions: Across datasets and input sizes, _most of the faithfulness is achieved through a small number of interactions_ (https://imgur.com/a/8sKgFrO).
> 2. Interaction signs: We explore the signs of the interactions for different sized inputs from _sentiment analysis_.(https://imgur.com/a/cyON8I2)
>
> **Fourier Intuition** Table 1 provides equations converting Fourier coefficients to Möbius transform, Shapley indices, and Banzhaf values. Fourier coefficients themselves are also interpretable as they are central to influence functions which are widely used for feature and data attribution. In particular, for subset $S$, with Fourier coefficient $F(S)$, the term $F(S)^2/|S|$ corresponds to the contribution of the interaction $S$ to the influence function of $i\in S$. We will add a paragraph in the camera-ready version.

---

> > ### Comment · Reviewer_e6K4 · 2025-04-06
> >
> > Overall speaking, the rebuttal is clear and addresses my concerns. The comparison to Kang et al. (2024) and the connection between Fourier transform and influence function are insightful and I believe they should be included in the main text, if accepted.
> >
> > Some further comments:
> >
> > 1.	Could you elaborate a bit more on “noise amplification of SMT in the transform domain”? In my view, it works as follows: the Mobius transform is $F(S)=\sum_{T\subseteq S} (-1)^{|T|-|S|} f(T)$, so for a $m$-th order transform, there are $2^m$ compositional terms, and this leads to an approximately exponential amplification of the noise in the original output $f(T)$. Is this understanding correct?
> >
> > 2.	The rebuttal clarifies that the paper mainly focuses on the Fourier transform rather than the Mobius transform (which is more widely-used by previous game-theoretic XAI literature). When validating the sparsity assumption, the paper refers to Kang et al. (2024) and Ren et al. (2024). However, the two papers demonstrate the sparsity of the Mobius transform (and an extension named AND-OR interactions) rather than the Fourier transform, thus making this validation unconvincing. Could you provide more evidence on the underlying sparse structure of the Fourier transform on deep neural networks?
> >
> > 3.	(Continued from previous question) How would you comment on the sparsity of Fourier transform and Mobius transform? Is the sparsity of one transform typically greater than the other?
> >
> > 4.	The paper shows the connection between the Mobius transform and the Fourier transform in Table 1. Then, given the efficient algorithm of computing the Fourier transform up to $n=1000$ input units in this paper, is it possible to scale up the computation of the Mobius transform to $n=1000$? If not, what is the main obstacle?
> >
> > 5.	This is a good work to scale up interaction-based methods, which are typically computational costly. I believe that opensourcing the code will lead to a more practical contribution.

---

> > > ### Author Response · Authors · 2025-04-07
> > >
> > > **Q1: Noise Amplification:** Your interpretation is a valid way of thinking about what we meant by noise amplification. This can be made a bit more rigorous as follows: Lets say $f(T)$  is a combination of a "true" sparse function $f'(T)$ and some noise $w(T) \sim \mathcal{N}(0,\sigma^2)$, i.i.d. for each $T$:
> > > $$f(T) = f'(T) + w(T), \quad w(T) \sim \mathcal{N}(0,\sigma^2) \text{ i.i.d.}$$ then the Fourier transform is:
> > > $$F(S) = F'(S) + W(S), \quad W(S) \sim \mathcal{N}(0,\sigma^2) \text{ i.i.d.}$$
> > > In contrast, the Möbius transform of $f$ denoted as $M$ has the following form:
> > > $$M(S) = M'(S) + W(S),\quad W(S)\sim \mathcal{N}(0,\sigma^22^{|S|}),\quad \mathbb{E}[W(S_1)W(S_2)] = \sigma^2 2^{|S_1\cap S_2|}$$
> > > This is of course, just an example, but shows that the Möbius transform can introduce correlated high-variance noise if the function deviates from an exactly sparse representation, and we do observe this type of behavior in practice. Real noise, however, is probably not i.i.d. gaussian (for example, it might depend on $|T|$). We think an interesting direction for future research in this space to further improve scaling is to study what type of noise is actually exhibited and build more tailored algorithms.
> > > Note this discussion is related to the concept of *whitening filters* in signal processing (the Fourier transform is a whitening filter for gaussian noise)
> > > https://en.wikipedia.org/wiki/Whitening_transformation.
> > >
> > > **Q2/Q3: Fourier vs. Möbius Sparsity:** We agree that we should have additional discussion beyond referencing Kang et al. (2024) and Ren et al. (2024) when discussing and justifying sparsity. We will develop this discussion with both a theoretical and empirical justification.
> > >
> > > *Theoretical Relationship (Möbius sparsity $\iff$ Fourier sparsity):* One argument we can use is that evidence of Möbius Sparsity implies Fourier sparsity. The sparsity of the Möbius transform ($s_M$) and Fourier transform $s_F$ are deeply connected. For example, a Fourier coefficient $F(S)$ results in at most $2^{|S|}$ Möbius coefficients. Since we are working with models that are also *low-degree* this means $|S|<d$ for some small $d$, (and typically most important $S$ have even lower $|S|$). Thus, we can upper bound the number of Möbius coefficients $s_M \leq 2^{d} \cdot s_F$, Conversely it is also true that $s_F \leq 2^{d} \cdot s_M$. **In practice, $s_M$ and $s_F$ are even closer together than the bound suggests** since (1) most of the important coefficients have $|S| < d$ and (2) the overlap between two Fourier interactions (conv. Möbius) $F(S_1)$ and $F(S_2)$ means they actually create only $2^{|S_1|} + 2^{|S_2|} - 2^{|S_1 \cap S_2|}$ coefficients. **Thus, by referencing Kang et al. (2024) and [1], which justify Möbius sparsity, this strongly supports the hypothesis of Fourier sparsity.**
> > >
> > > *Empirical Evidence:* For empirical evidence of Fourier sparsity, we can refer to this figure (https://imgur.com/a/8sKgFrO) which we provided in our rebuttal, and will include in our camera-ready version. Furthermore, Fig. 5 also serves as a justification since achieving high faithfulness with SPEX shows that these deep neural networks based models are well represented by a sparse Fourier transform. Comparing the full Fourier vs Möbius transforms on the first 9 movie reviews from *sentiment*, we find that the Fourier transform produces a much sharper decay in the sorted magnitudes of coefficients (https://imgur.com/a/UDkAdsf); however, this observation may be task or model dependent, so a more thorough comparison of the sparsity of the two transforms is another interesting future research direction.
> > >
> > >
> > > **Using SPEX to Compute Möbius and Other Interactions** Yes. Since it is very easy to convert from the Fourier to the Möbius transform, SPEX solves the problem of computing the Möbius transform at very large scales efficiently. Furthermore, since the Möbius transform is so deeply related to many of the other interaction indices (Shapley-Taylor, Faith-Shapley, etc.) it is also very easy to compute these other interaction indices once we have learned a Fourier transform.
> > >
> > > **Public and Easy-to-Use Code** Our code will certainly be made available upon publication. We have put significant effort into making it easy to use for practitioners and researchers.  Our goal is to push forward the literature on feature attribution to be in line with the scale of the best models available today. Recently Llama-4-Scout released with a 10M token context limit, so this is a very timely problem!
> > >
> > > ***We thank you again for your time. We hope you will consider this discussion and our thorough answers to your other questions in your final revision.***
> > >
> > > [1] Ren, Qihan, et al. "Where we have arrived in proving the emergence of sparse symbolic concepts in ai models." (Our understanding is that this work's proof is about Möbius sparsity, and not the AND-OR extension, which are studied in other works by overlapping authors).

---

### Official Review · Reviewer_KaFj · 2025-03-11

**Overall Recommendation:** 4

**Summary:**

The paper introduces **SPEX**, a scalable method for explaining LLM predictions by recovering feature interactions using **structured feature masking (BCH codes) and sparse Fourier recovery**. SPEX efficiently identifies the most important feature interactions without evaluating all \( 2^n \) subsets, making it the first method to scale to long-text inputs (1000+ features). It outperforms SHAP and Faith-Banzhaf in scalability while maintaining high faithfulness \( R^2 \).

**Claims And Evidence:**

The paper claims SPEX provides **efficient, scalable feature interaction recovery** and **highly faithful explanations**. While experiments support efficiency claims, faithfulness could be inflated by **memorization bias**, where LLMs correctly predict masked inputs due to seen training data. An evaluation on **unseen data** would better validate this claim.

**Essential References Not Discussed:**

No

**Experimental Designs Or Analyses:**

The experiments demonstrate SPEX’s scalability and faithfulness but lack **tests on out-of-distribution (OOD) inputs**. Additionally, the complexity of solving for \( c_T \) in BCH decoding is not fully analyzed, potentially hiding computational overhead.

**Methods And Evaluation Criteria:**

SPEX’s structured masking and sparse Fourier transform effectively reduce computational complexity, making it suitable for LLMs. However, its reliance on **faithfulness metrics** may not fully capture causal attributions. Adversarial masking techniques (e.g., synonym swaps) could strengthen evaluation.

**Other Comments Or Suggestions:**

No

**Other Strengths And Weaknesses:**

_+_ Efficient and scalable feature interaction explanations.
_-_ Faithfulness may be inflated due to memorization bias.
_-_ Does not analyze decoding complexity in detail.

**Questions For Authors:**

1. How does SPEX perform on unseen (OOD) data? Would faithfulness metrics hold without memorization effects?
2. Would synonym swaps help detect spurious feature attributions?
3. What is the computational complexity of BCH decoding? Are there efficiency trade-offs at scale?

**Relation To Broader Scientific Literature:**

The paper relates well to work on **SHAP, Faith-Banzhaf, and interaction-based feature attributions**.

**Theoretical Claims:**

The Fourier-based interaction recovery assumes **feature independence**, which may not hold for LLMs that rely on **context-dependent reasoning**.

---

> ### Author Rebuttal · Authors · 2025-04-01
>
> **This rebuttal contains (anonymized) links to figures. We also built a web app to help explore SPEX: https://anon858023.github.io/spex-webapp/**
>
> Thank you for your constructive and positive review! We appreciate your insightful comments and have addressed them with additional experiments and clarifications below. We believe that SPEX's unique scalability for computing feature interactions in long-text inputs represents a significant advancement beyond the state of the art, and we hope this will be reflected in your final evaluation.
>
> **Memorization** This is an interesting hypothesis. Memorization could lead to the phenomenon you suggest, where only a few tokens suffice to determine the full sequence. Note that all baseline methods are masking based, and thus our SOTA performance claim is not invalidated, even if this hypothesis is valid. In practice, however, we find evidence that this hypothesis does not hold in our datasets:
>
> 1.  **Variance**: We find that the variance on the test dataset we use to evaluate faithfulness is not small. That is, depending on the masks, the model output changes significantly. We visualize these model outputs for examples from the *Sentiment* dataset
> (https://imgur.com/a/pagr97R). This is further validated by Fig. 4, which show that marginal attributions, which perform a linear fit, are unable to capture the complex nature of the model output.
>
> 2. **Explicitly avoid memorization bias** An evaluation on unseen data is difficult for the models used in our paper since we do not know the pre-training data that was used. However, to address this concern, we repeat our faithfulness experiments on the TriviaQA dataset for OLMo2-7B-Instruct, an open-source/open-data language model. TriviaQA was not included in the pre-training corpus and was also not used at all during model development for OLMo2-7B-Instruct.  Due to time constraints, we do not consider every regime of $n$, and only compare to Faith-Banzhaf since it was strongest in our previous experiments.
>
> | Model                 | n         | SPEX   | Banzhaf | Faith-Banzhaf 2nd |
> | :-------------------- | :-------- | :----- | :------ | :---------------- |
> | *OLMo2-7B-Instruct* | 32--63    | **0.761** | 0.410   | 0.632             |
> |                       | 64--127   | **0.651** | 0.378   | ---               |
> |                       | 128--255  | **0.589** | 0.340   | ---               |
>
> *Table: Faithfulness on TriviaQA dataset (held-out on OLMo2-7B-Instruct) across input lengths ($n$) for OLMo2-7B-Instruct*
>
> **Theoretical Claims** (*We tried our best to answer based on what you wrote, but, please feel free to clarify if we did not answer the question here, as there are many ways to interpret your point about feature independence*) The boolean Fourier transform is defined under a Hilbert space of functions with inner product $\langle f,g \rangle = \mathbb{E}_{\mathbf{x} \sim B^n} f(\mathbf{x})g(\mathbf{x})$. The induced distance metric essentially measures the average distance ($\ell_2$) in output between the two functions when features are i.i.d Bernoulli(1/2). However, the transform itself remains well-defined and interpretable even with feature dependencies. Our faithfulness metric (only one of our evaluation criteria) is related to the $\ell_2$ distance in the output space, provides a valuable measure of how well our recovered interactions approximate the original model's behavior. While other metrics might be more relevant for correlated features, the optimal metric for interpretability remains an open question. Importantly, SPEX can effectively capture learned correlations within the LLM.
>
> *Example*: When the LLM has already learned some of the correlations in the data, SPEX can extract this. For example, when two words ($w_1$ and $w_2$) always appear together, the model will not be impacted when either $w_1$ or $w_2$ are masked (since seeing either $w_1$ or $w_2$ is enough to deduce the presence of the other), and will only change when both $w_1$ and $w_2$ are masked simultaneously.  SPEX will find a $2^{nd}$ order interaction here that indicates this correlation.
>
> **Synonym Swap** We appreciate this suggestion. While valuable, introducing synonym swaps adds potential confounding factors and poses scalability challenges for comprehensive evaluation. We believe this direction warrants further investigation in future work.
>
> **BCH Complexity** The complexity of decoding a $BCH(n_c, k_c, t)$ code is in Appendix A.5. It is $O(n_ct + t^2)$, with $n_c \approx n + t\log(n)$. Soft-decoding offers a way to trade off complexity for superior performance. We implement a 2-chase decoder which has an additional $2^{d_{chase}}$ factor, where $d_{chase} \leq t$ can be designed. We emphasize that this complexity is small compared to current interaction indices which have complexity that scales as $O(n^d)$, where $d$ is the highest degree of interaction considered. We will include a discussion in the final revision.

---

### Decision · Program_Chairs · 2025-05-01

**Decision:**

Accept (poster)

**Comment:**

This paper presents SPEX, a scalable feature interaction explanation method based on sparse Fourier transforms and channel decoding, enabling interaction attributions for LLMs with very large input spaces (~1000 tokens). The reviewers appreciated the significance of the problem: interaction-based explanations at scale, and agreed that the algorithmic contributions, experiments, and theoretical insights represent a meaningful advancement over previous work like Kang et al. (2024). The authors provided thorough and convincing rebuttals, clarifying the key differences from related sparse transform approaches, discussing memorization bias, out-of-distribution generalization, and robustness to noise. Additional evaluations on new models (Qwen2, Mistral) and robustness experiments were added, further strengthening the submission. Some concerns about deeper connections between Fourier and Möbius sparsity, and future extensions into mechanistic interpretability, were discussed in detail. Overall, this is a strong paper that tackles an important bottleneck in explainability research for LLMs with rigorous experiments and theoretical framing.